# The Epigenetics of Migraine

**DOI:** 10.3390/ijms24119127

**Published:** 2023-05-23

**Authors:** Farzin Zobdeh, Ivan I. Eremenko, Mikail A. Akan, Vadim V. Tarasov, Vladimir N. Chubarev, Helgi B. Schiöth, Jessica Mwinyi

**Affiliations:** 1Department of Surgical Sciences, Functional Pharmacology and Neuroscience, Uppsala University, Husargatan 3, P.O. Box 593, 75124 Uppsala, Sweden; zobdeh_farzin@yahoo.com (F.Z.); eremenko_i_i@student.sechenov.ru (I.I.E.); akan.mikhail@mail.ru (M.A.A.); helgi.schioth@neuro.uu.se (H.B.S.); 2Advanced Molecular Technology, LLC, 354340 Moscow, Russia; tarasov-v-v@mail.ru (V.V.T.); tchoubarov@mail.ru (V.N.C.)

**Keywords:** migraine, epigenetics, microRNA, circRNA, DNA methylation, histone acetylation

## Abstract

Migraine is a complex neurological disorder and a major cause of disability. A wide range of different drug classes such as triptans, antidepressants, anticonvulsants, analgesics, and beta-blockers are used in acute and preventive migraine therapy. Despite a considerable progress in the development of novel and targeted therapeutic interventions during recent years, e.g., drugs that inhibit the calcitonin gene-related peptide (CGRP) pathway, therapy success rates are still unsatisfactory. The diversity of drug classes used in migraine therapy partly reflects the limited perception of migraine pathophysiology. Genetics seems to explain only to a minor extent the susceptibility and pathophysiological aspects of migraine. While the role of genetics in migraine has been extensively studied in the past, the interest in studying the role of gene regulatory mechanisms in migraine pathophysiology is recently evolving. A better understanding of the causes and consequences of migraine-associated epigenetic changes could help to better understand migraine risk, pathogenesis, development, course, diagnosis, and prognosis. Additionally, it could be a promising avenue to discover new therapeutic targets for migraine treatment and monitoring. In this review, we summarize the state of the art regarding epigenetic findings in relation to migraine pathogenesis and potential therapeutic targets, with a focus on DNA methylation, histone acetylation, and microRNA-dependent regulation. Several genes and their methylation patterns such as *CALCA* (migraine symptoms and age of migraine onset), *RAMP1*, *NPTX2*, and *SH2D5* (migraine chronification) and microRNA molecules such as miR-34a-5p and miR-382-5p (treatment response) seem especially worthy of further study regarding their role in migraine pathogenesis, course, and therapy. Additionally, changes in genes including *COMT*, *GIT2*, *ZNF234*, and *SOCS1* have been linked to migraine progression to medication overuse headache (MOH), and several microRNA molecules such as let-7a-5p, let-7b-5p, let-7f-5p, miR-155, miR-126, let-7g, hsa-miR-34a-5p, hsa-miR-375, miR-181a, let-7b, miR-22, and miR-155-5p have been implicated with migraine pathophysiology. Epigenetic changes could be a potential tool for a better understanding of migraine pathophysiology and the identification of new therapeutic possibilities. However, further studies with larger sample sizes are needed to verify these early findings and to be able to establish epigenetic targets as disease predictors or therapeutic targets.

## 1. Introduction

Migraine is one of the most common types of primary headache disorders, which affects over one billion people globally with a female predominance. Migraine is a major cause of disability, which obtrudes an immense socioeconomic burden [1]. Migraine can be divided into different types on the basis of the neurosymptomatic picture, including major types such as migraine with and without aura, as well as less frequently occurring types such as ocular, abdominal, vestibular, and hemiplegic migraine. According to the frequency of attacks, migraine can be divided into chronic and episodic [2]. 

The precise pathophysiology of migraine is not fully understood. Despite the noticeable therapeutic progress in recent years, migraine treatment is often unsatisfying [3]. Triptans (e.g., sumatriptan and zolmitriptan) and nonsteroidal anti-inflammatory drugs (e.g., ibuprofen) have been the leading options for treating acute migraines for many years. Additionally, the most recent therapeutic agents such as gepants (e.g., rimegepant) and ditans (lasmiditan) are considered promising options for the treatment of acute migraine [4,5]. For preventive treatment, different drug classes including beta-blockers (e.g., propranolol), tricyclic antidepressants (e.g., amitriptyline), and anticonvulsants (e.g., topiramate) have been used. Furthermore, novel drugs which inhibit calcitonin gene-related peptide (CGRP) or its receptor as validated targets for migraine therapy have been shown to be efficient in migraine preventive treatment. Examples are monoclonal antibodies which block either CGRP (e.g., galcanezumab) or CGRP receptor (e.g., erenumab) [5,6]. The diversity of drug classes used for acute and preventive treatment with various mechanisms of action reflects the overall still limited understanding of migraine pathophysiology. According to the current pathophysiological understanding, migraine attacks are divided into four phases, i.e., the premonitory phase, the aura phase, the headache phase, and the postdrome phase [7,8]. The premonitory phase usually starts 48–72 h before the onset of a migraine attack [2,9] with common symptoms such as fatigue, neck stiffness, photophobia, yawning, food craving, and concentration difficulties, and it is characterized by hypothalamic activation and an increased parasympathetic tone [8]. The aura phase occurs in almost 30% of migraine patients [10]. Although the precise pathophysiological mechanism behind the aura is not fully understood, to date, cortical spreading depression (CSD) is considered to be the leading cause of aura [11]. CSD is an abnormal event that is characterized by a slowly propagating wave of depolarization of cortical neuronal and glial cells, followed by a depression of electrical activity [12]. CSD is correlated with a massive influx of sodium, calcium, and water, along with an efflux of potassium, proton, glutamate, ATP, and neurotransmitters, which may link aura to head pain via the activation of perivascular trigeminal nerve endings [13]. The headache phase is characterized by the activation of the trigeminovascular system and sensory transmission of nociceptive signals, which results in the release of neurotransmitters such as CGRP, pituitary adenylate cyclase-activating polypeptide-38, glutamate, and nitric oxide, as well as the activation of vascular and meningeal nociceptors [5,14]. The postdrome phase after the headache attack can include neuropsychiatric (e.g., anxiety, irritability, and yawning), sensory (phonophobia, photophobia, focusing difficulty, speech difficulty, and hypersensitivity), gastrointestinal, and other symptoms (asthenia and weakness), which can last a day [15].

A correlation between hereditary elements and migraine has been studied since the 1990s [16]. Studies could demonstrate that interactions between genes and the environment are a major cause of migraine [17,18]. Migraine is principally thought to be of polygenic nature. However, monogenic forms are known to occur in rare cases such as, e.g., familial hemiplegic migraine, with mutations in genes such as *CACNA1A*, *ATP1A2*, and *SCN1A* [19,20]. Another example involves mutations in genes such as *PNKD*, *SLC2A1*, *SLC1A3*, *SLC4A4*, and *PRRT2* that may occur in hemiplegic migraine [21]. Moreover, mutations in *KCNK18* were found to be related to a monogenic form of typical migraine with aura [22]. Even though genetics plays a considerable role in migraine, the exact mechanisms that underly the observed genotype–phenotype associations have not yet been elucidated in studies [23]. 

Migraine is not a static disorder but rather dynamic. Duration of attack, severity, frequency, and symptom composition of attacks can change over time. A further understanding of the molecular-biological mechanisms regulating these events is of major importance. Of note, the number of studies investigating the role of epigenetic mechanisms in migraine is currently growing. Recently, epigenetics has been the focus of several animal and human studies, and it appears to be a promising approach to further explain migraine pathogenesis, course, and severity, as well as the high variability in migraine-associated therapy response [24,25,26]. Epigenetic mechanisms regulate cell-cycle development by controlling the expression of individual genes [27]. DNA methylation and histone modifications are epigenetics mechanisms with profound effects on the regulation of gene expression [28,29]. DNA and histone modifications regulate DNA expression without changing the DNA sequence and can be heritable or of inducible nature [25,30,31]. Importantly, epigenetic mechanisms such as methylation or histone acetylation transmit environmental signals to the cells, which leads to modifications of the functional output of the cell genome [32]. Therefore, environmental factors may trigger migraine as a result of changes in gene expression [33]. Likewise, studies have started to investigate the role of microRNAs in migraine headache [26,34]. microRNAs are small endogenous RNAs that regulate gene expression post-transcriptionally [35]. To date, 30–60% of protein-coding genes in mammals are anticipated to be regulated by microRNAs [36,37]. The interaction between microRNAs and target mRNAs results in expression-repressive effects [38]. The expression of microRNAs is dynamic and, therefore, worth studying in relation to migraine due to its dynamic nature regarding attacks and severity that may change throughout life [39,40]. In this review, we shed light on the current knowledge regarding the role of DNA methylation, histone acetylation, and microRNA shifts in migraine pathophysiology and their potential as future therapeutic predictors or targets which might be considered valuable for further research.

## 2. Methodology

### 2.1. Literature and Search Strategy

The search was conducted in the databases PubMed, Scopus, Google Scholar, and Cochrane Library, using the following search terms in different combinations and different long combination chains: (“epigenetics” OR “epigenomics”) AND (“migraine”) AND (“DNA methylation” OR “histone acetylation” OR “histone methylation” OR “circRNA” OR “microRNA”).

### 2.2. Inclusion and Exclusion Criteria

Selected articles were required to meet the following criteria:(1)The study contained original data.(2)The study was in vitro or in vivo.(3)The study subjects were human or animal.(4)The study was written in English.(5)The study described the interplay between the migraine pathogenesis or potential targets for obtaining effective therapeutic responses and epigenetics.

### 2.3. Selected Studies

A total of 27 articles were chosen according to the abovementioned criteria to be presented in this review (Table 1 and Table 2).

## 3. Principal Mechanisms of DNA Methylation and Histone Modifications

Human DNA is packaged in an eminently organized chromatin, with a core formed by five families of histones that form the basis of chromatin plasticity. Histone post-translational covalent modifications, such as acetylation, methylation, phosphorylation, sumoylation, ubiquitination, and other mechanisms, are an important part of gene regulation that is referred to as the histone code. The histone code determines the accessibility of a DNA fragment to RNAs and transcriptional proteins. Histone modifications are settled by histone-modifying enzymes and are a key to dynamic and long-term epigenetic regulation of DNA replication, transcription, and repair [63,64]. In contrast, DNA methylation may reduce gene expression by directly impairing the binding of transcriptional activators or by the recruitment of methyl-binding proteins, which subsequently allows the recruitment of transcriptional corepressor complexes. Mentioned mechanisms of epigenetic regulation directly cooperate through multiple pathways and effectively regulate gene expression [65].

### 3.1. DNA Methylation and Demethylation

DNA methylation is the main form of epigenetic modification and is achieved by the addition of a methyl group to the 5′-cytosine of CpG groups. This chemical reaction is catalyzed by DNA methyl transferases (DNMTs), i.e., DNMT1, DNMT3A, and DNMT3B, using S-adenosylmethionine as the methyl group donor [66]. The resulting 5-methyl cytosine promotes a closed chromatin conformation, decreasing transcription of a specific DNA region and, thus, leading to regulatory changes in many biological processes [63,64,67,68]. While DNMT1 is involved in DNA maintenance methylation and acts on the nonmethylated strand of hemimethylated DNA, DNMT3A and DNMT3B are de novo methyltransferases that methylate both strands of unmethylated DNA [69,70]. Accordingly, DNMT1 plays a significant role as a maintenance methyltransferase responsible for the preservation of DNA methylation after DNA replication. In contrast, DNMT3 is involved in establishing DNA methylation patterns during embryonic development and in their change during life. Given the importance of de novo methyltransferases in mammalian development, DNMT3 proteins could be implicated in virtually all diseases. Polymorphisms of the DNMT3A and DNMT3B genes may alter gene expression and affect their enzymatic activity, and they have been shown to contribute to a variety of pathological conditions including lung cancer, colorectal cancer, prostate cancer, myelodysplastic syndrome and other hematological disorders, Alzheimer’s disease, Parkinson’s disease, and immunodeficiency [65,71,72]. The methylation of cytosine in the human genome occurs mainly within a 5-’CpG-30 (CpG) dinucleotide, yet non-CpG methylation with an uncertain role in mouse and human embryonic stem cells has also been described [68]. Regions of the human genome that have an atypically high frequency of CpG occurrence compared to the rest of the genome are called CpG islands (CGIs). About half of all CGIs are found at gene promoter sites, thus affecting the transcription of particular genes [73]. However, the other half are distributed between sites within gene bodies (intragenic) or between genes (intergenic) and are termed “orphan” CGIs [68]. Intragenic CGIs (iCGIs) can impact gene expression in a variety of ways; they have been described to participate in tissue-specific DNA methylation, implying that the regulation of iCGIs is crucial for tissue-specific programming, such as in the brain [74,75]. In turn, iCGIs may have a potential function as tissue-specific alternative promoters for downstream genes [76]; however, the biological significance of these “orphan” CGIs remains largely elusive. DNA demethylation is mediated by the ten-eleven translocation (TET) proteins TET1, TET2, and TET3 [77]. TET proteins cause the oxidation of 5-methyl cytosine to 5-hydroxymethylcytosine, 5-formylcytosine, and 5-carboxylcytosine [78,79]. The oxidized intermediates, 5-formylcytosine and 5-carboxylcytosine, are then lost in subsequent DNA replication or removed by thymine DNA glycosylases and base excision repair proteins, thus regenerating unmethylated cytosines at targeted sites [64,80,81]. The expression of the DNMTs and TETs varies depending on the tissue and is generally high in the brain, where DNA methylation and demethylation processes are crucial for the normal development and functioning of the brain, such as the proliferation and differentiation of neural stem cells, neuronal activity, and synaptic plasticity [67,78,81,82,83]. Altered DNA methylation is found to be involved in various pathologies such as cancer [84,85] and neuropsychiatric diseases including schizophrenia, major depressive disorders, Alzheimer’s disease, and Parkinson’s disease, suggesting its role in migraine [86,87,88,89,90].

### 3.2. Histone Acetylation and Deacetylation

Histone acetylation is crucial for active gene transcription to influence the compaction state of chromatin by neutralizing positive charges of histones and decreasing the electrostatic interaction between negatively charged DNA and histones. This allows the DNA to unwind, making it more accessible to proteins that regulate transcription [63,91,92]. Although the significance of each modification of histone acetylation has not been fully clarified, many studies have shown that histone acetylation has a crucial role in the regulation of gene expression. Thus, alteration of histone acetylation patterns may be involved in virtually all biological processes.

#### 3.2.1. The Principle of Histone Acetylation

Histone acetylation occurs when the acetyl group, using acetyl coenzyme A as a donor, is added to the ε-amino group on lysine residues at the N-terminus of histone. These reactions are catalyzed by histone acetyltransferase (HAT) enzymes and lead to a reduced positive charge of the histone. The opposite reaction, i.e., histone deacetylation, is catalyzed by histone deacetylases (HDACs), a complex family of proteins responsible for removing the acetyl group from ε-N-acetyl lysine residues that are added by histone acetyltransferases. This, thus, restores the positive charge of the histone [91,93]. To date, 18 different HDACs have been characterized and are classified into four major classes (I, II, III, and IV) [93,94,95].

#### 3.2.2. Histone Acetylation and Cancer

Multiple studies have demonstrated that aberrant histone acetylation modifications are related to cancer in general, including breast cancer. Lower expression of HAT1 is associated with the pathogenesis of lung cancer, while it was highly expressed in hepatocellular carcinoma, nasopharyngeal cancer, and pancreatic cancer, acting as an oncogene, associated with poor prognosis [96]. In addition, aberrant expression of HDACs has been linked to a variety of malignancies, including solid and hematological tumors. HDAC inhibitors (HDACis) have been demonstrated to induce considerable therapeutic effects in various cancers [97,98].

#### 3.2.3. Histone Acetylation and Immune Response

The HDAC/HAT ratio regulates the expression of several genes including those involved in inflammatory diseases. Type A HATs, such as CBP and p300, regulate the pathway mediated by nuclear factor kappa-light-chain-enhancer of activated B cells (NF-κB), which is responsible for modulating inflammatory response [96]. Additionally, levels of different HDACs have been shown to alter significantly in patients with inflammatory diseases such as rheumatoid arthritis and chronic obstructive pulmonary disease, further suggesting a crucial role of histone acetylation patterns in immune response [99].

#### 3.2.4. Histone Acetylation and Embryonic Development

Histone acetylation is one of the primary mechanisms causing early programming of cell proliferation and differentiation. During development, multiple waves of epigenetic changes take place. The first wave occurs after fertilization and leads to zygotic gene activation. The next wave occurs during blastocyst formation, during which histone modifications are altered, and both X chromosomes are reactivated in female cells. After implantation, another wave of epigenetic programming takes place, in which chromatin accessibility becomes progressively decreased [100]. Class I HDACs are expressed in developing and adult brains, pointing to their role in brain development and function. HDAC1 plays a significant role in neurotoxicity by affecting axon transport and mitochondrial activity; thus, it might be associated with neurodegenerative processes [101,102,103,104,105]. HDAC2 is crucial for the regulation of brain tissue maturation. Inhibition of HDAC2 leads to cell death in many cells and HDAC2 knockout affects microglia maturation. HDAC2 is also involved in cognitive impairment, e.g., Alzheimer’s disease [104,105,106,107]. In class II HDACs, HDAC4 has been shown to affect synaptic plasticity in mice [104,108,109]. The data mentioned above show the strong impact of histone acetylation on numerous biological processes, including brain tissue function and the development of neurological disorders, which may imply a role in migraine development.

#### 3.2.5. Histone Deacetylase Inhibitors as Therapeutic Agents in Neurologic Disorders

HDACis are a heterogeneous group of agents that inhibit HDAC activity, as well as affect the acetylation state of non-histone proteins, such as p53, Hsp90, STAT3, and NF-κB, regulating the stability and/or DNA-binding properties of these non-histones. To date, many selective and multitarget HDACis have been developed, and some of them have been approved for the treatment of cancer [110,111]. Additionally, their anticancer activity has been reported widely in the literature, including colorectal, hepatocellular, pancreatic, breast, thyroid, lung, and endometrial cancers. In addition, HDACis appear to be also crucial for the therapy of other diseases, including neurological disorders, in which epigenetic dysregulation may play an important role in disease development. Various HDACis show a positive effect in models of Alzheimer’s disease, Parkinson’s disease, Huntington’s disease, amyotrophic lateral sclerosis, and Friedrich ataxia [112]. Despite the considerable role of HDACis in different neurological disorders, only limited data on the use of HDACis in migraine models is available. A recent study on rat models of medication overuse headache showed that two HDACis (panobinostat and givinostat) could counteract medication overuse headache (MOH) symptoms. Treatment with panobinostat or givinostat prevented overexpression of CGRP and its receptor, key components in migraine development, in the trigeminal ganglion (TG) of the MOH model. Furthermore, both drugs prevented dermal vasodilation and photophobic behavior, as well as partly prevented craniofacial allodynia [113]. Such promising data may imply that HDACis may be used as a therapeutic agent in migraine patients in the future. However, further investigations on this topic are needed in migraine.

### 3.3. Histone Methylation and Demethylation

Histone methylation usually occurs at the arginine or lysine N-terminal region, which leads to the activation or inhibition of gene expression. Each lysine residue can be mono-, di-, or trimethylated on the ε-amino group of lysine, while arginine can be monomethylated or dimethylated symmetrically or asymmetrically. Histone methylations are catalyzed by histone methyltransferase (HMT) enzymes, which can add a methyl group from S-adenosylmethionine to their target residue. Currently, HMTs are classified into three families, which include the SET domain-containing enzymes and Dot1-like proteins that act on lysines. The third family consists of arginine N-methyltransferase enzymes, which methylate arginines. Histone demethylases are enzymes that remove the various methyl groups from lysine or arginine [114].

Like other epigenetic modifications, histone methylation and demethylation exhibit a complex regulation of gene expression, and a change in these processes may play a role in the development of various diseases, including neurological disorders such as Alzheimer’s disease, Huntington’s disease, Parkinson’s disease, and amyotrophic lateral sclerosis [115]. Despite the strong association between histone methylation and the mentioned neurological disorders, there are no data available on its association with migraine.

## 4. Environment, Epigenetics, and Migraine

The epigenome can be altered by environmental factors such as stress, diet, and toxicants. A changed epigenome can influence growth, development, and disease risk. The interplay between the genetic background and the epigenetic environment seems to be more relevant during critical developmental periods, such as in childhood, where the epigenome shows elevated plasticity. Hundreds of studies in human cohorts and animal models have shown associations between epigenetic changes in the offspring and the gestational environment, e.g., gestational exposures to toxicants [116] and maternal stress [117]. Most studies to date have focused on DNA methylation; however, the effects of environmental factors on other epigenetic modifications such as histone modifications and miRNA expression are also emerging.

### 4.1. Stress

Responses to stress comprise a complex interplay of molecular, hormonal, neuronal, and behavioral processes. These adaptations operate on multiple levels and are necessary for effectively coping with stressors and the maintenance of a physiological and cellular balance. One of the principal effectors of the stress response is the hypothalamic–pituitary–adrenal axis. The hypothalamic–pituitary–adrenal axis is regulated by the hypothalamus, a region of the brain that signals the anterior pituitary to secrete adrenocorticotropic hormone, which then drives the adrenal release of glucocorticoids in blood. Glucocorticoids primarily exert their actions in target tissues by activating two receptors, the mineralocorticoid receptor and the glucocorticoid receptor (GR). GR acts as a transcription factor that regulates gene transcription by binding as a dimer to glucocorticoid response elements [118,119,120]. Notably, glucocorticoid response element binding has been shown to not only regulate gene transcription but also elicit persistent changes in DNA methylation and demethylation, both at the genome-wide level and within selective gene loci. Furthermore, glucocorticoids can induce histone modifications, through direct GR binding or via interaction of GRs with other transcription factors that recruit HATs [121]. Several studies reported an association of stress-driven epigenetic changes with stress-related disorders such as post-traumatic stress disorder (PTSD), depression, and various psychotic disorders, which may allow the hypothesis of a putative role in migraine development. However, to date, no article has explored this association in migraine patients.

### 4.2. Diet

A diet-modifying strategy has been proposed as a potential treatment of various diseases, including migraine, as certain dietary compounds with specific mechanisms of action can potentially interfere with disease pathogenesis. This “epigenetic” diet may be able to alter the epigenetic profile of consumers with specific conditions, thus preventing such conditions. However, the underlying mechanisms of such modifications at the molecular level of the epigenetic profile remain unclear. In the context of migraine, where the impact of epigenetic influences on the disease has gained considerable attention, such a dietary intervention would aim to oppose epigenetic mechanisms underlying migraine or to promote preventive mechanisms. Folate, which is involved in DNA methylation and which has been shown to be beneficial in migraine, has captured further attention in the context of an epigenetic diet for migraine [64]. It has been proposed that defining a diet that can target DNA methylation, such as a diet rich in folate, could be a promising avenue for future investigations of epigenetic dietary factors associated with migraine [122]. However, further research is necessary to provide evidence on the dietary components that could interfere with the epigenetics of migraine.

### 4.3. Toxins

Exposure to environmental toxins, especially in childhood, has also been associated with epigenetic alterations. Specifically, alterations in DNA methylation have been associated with exposure to bisphenol A, polycyclic aromatic hydrocarbons, phthalates, high levels of nitrogen dioxide, and fine particulate matter [123]. A recent study performed by Guo et al. [124] found one CpG site (cg27510182) on the gene *DAB1* that potentially mediates the effect of polycyclic aromatic hydrocarbons on social problems in children. However, the precise contribution of epigenetic changes caused by the mentioned compounds in the development of brain disease remains unclear. Of note, increasing evidence also shows that early-life metal exposure, such as cadmium, lead, methylmercury [125], and arsenic [126], may modulate the epigenetic landscape in the brain. In combination with metal-induced neurotoxicity, these epigenetic changes might contribute to the development of brain disease and affect neurodevelopment outcomes such as psychomotor development index and rating scale of emotional regulation; however, their association with neurological disorders remains to be investigated [127]. Experimental, clinical, and epidemiological research has provided evidence that pesticide exposure, even at low levels, has long-term effects on the central nervous system and is able to induce epigenetic changes, e.g., alteration of global DNA methylation or miRNA expression [128]. However, the impact of such epigenetic changes on the development of neurological disorders such as migraine remains unclear.

The mentioned environmental factors contribute to epigenetic changes, which in turn may, in a complex manner, play a role in the development of complex neurological disorders such as migraine. Therefore, further comprehensive research is necessary to explore the intricate relationship between epigenetics and the environment.

## 5. Aberrant DNA Methylation Patterns in Migraine

DNA methylation is the most common type of epigenetic modification and plays a key role in several disorders including cancer, vascular, neurodegenerative disorders, and migraine [33,129,130]. Aberrant epigenetic patterns can be used as biomarkers for the diagnosis and prognosis of many diseases and may even be capable of distinguishing different subtypes of a specific disease [131,132,133,134]. It should be acknowledged that DNA methylation may differ between tissues. Thus, ideally, methylation should be studied in the tissue of interest. In the case of migraine, brain tissue would be the ideal tissue to be used, which is, however, impossible in humans. Hence, blood is used to analyze DNA methylation patterns of leukocytes in humans. Labruijere et al. compared the methylation levels of calcitonin-related peptide alpha (*Calca*), receptor activity-modifying protein 1 (*Ramp1*), calcitonin receptor component protein, calcitonin receptor-like receptor, upstream stimulating factor 2, estrogen receptor 1, G-protein-coupled estrogen receptor 1, nitric oxide synthase 3, and methylenetetrahydrofolate reductase (*Mthfr*) genes between migraine-related tissues (dura mater, TG, and trigeminal caudal nucleus) and peripheral control tissues (aorta and leukocytes) in rats [41]. However, no correlation was shown between DNA methylation in the leukocyte samples and samples from other tissues for any of the genes. In addition, Labruijere et al. investigated the concordance of DNA methylation in human leukocytes and rat leukocytes. From a sample of 395 healthy women, comparable values of DNA methylation in the genes of interest in human and rat leukocytes were observed [41]. On the basis of observed data, it may be concluded that rat leukocytes are not representative of changes in DNA methylation in the trigeminal caudal nucleus and TG.

Bainomugisa et al. assessed the overlap between the DNA methylation of 1036 genes (1453 CpGs) associated with PTSD and genes associated with migraine in 15 pairs of monozygotic twins, discordant for migraine. Of those, DNA methylation of 99 genes (132 CpGs) associated with PTSD was also associated with migraine [42]. In this study, 62 genes were also investigated, previously identified by Gerring et al., which were differently methylated in migraineurs compared to healthy controls (HC), as detected in a sample of 67 migraineurs and 67 healthy controls [43]. In the migraine monozygotic twins, 46 out of 62 genes contained at least one CpG site that was significantly associated with migraine. Of those, six genes, *KCNG2*, *DGKG*, *SND1*, *LHX6*, *ADIRF*, and *RPTOR*, survived multiple testing correction in the study [42,43]. Interestingly, in the study performed by Gerring et al., *DGKG* was shown to be the fourth most significant differentially methylated region associated with migraine. *DGKG* encodes diacylglycerol kinase gamma, which is highly expressed in rat brain, particularly the cerebral cortex, hippocampal formation, and cerebellum, suggesting a physiological importance of this enzyme for proper brain function [43].

## 6. Migraine Chronification and DNA Methylation

In a number of patients, EM may progress to chronic migraine, a migraine form that accounts for ~8% of the total migraine population [135,136]. Factors that have been identified to predispose to the chronification of migraine include genetic variation and epigenetic changes, overuse of acute headache medications, and a high baseline attack frequency [137]. Environmental factors are thought to play an important role in migraine. These factors may directly induce an acute migraine attack and may lead to epigenome alteration. Epigenetic regulation is described to play a role in chronic changes in the brain tissue, which may also predispose to migraine development. Although the exact biological mechanisms that lead to the transition from episodic to chronic headache are unknown, it has been hypothesized that frequent headache attacks may lower the threshold for subsequent headache attacks through epigenetic mechanisms, resulting in a feedforward loop [33].

One epigenome-wide retrospective case–control study covering 11 years aimed to investigate the transformation from episodic to chronic headaches, in a mixed sample, which also included migraineurs. Winsvold et al. assessed the methylation level at 485,000 CpG sites in a total of 36 female headache patients, who transformed from episodic to chronic headache, matched with 35 female patients with headache who did not progress to chronic headache [44]. None of the top 20 identified CpG sites associated with the chronification of headache reached statistical significance after multiple testing correction. In subsequent combined statistical analyses of identified CpG sites, it was shown that the two most strongly associated CpG sites were related to *SH2D5* and *NPTX2* genes. However, obtained results may be explained by the mixed sample, as approximately half of the chronic headache patients fulfilled the criteria for chronic migraine, likely contributing to the observed data; hence, a study with a bigger sample size and distinction of chronic migraine is needed in order to detect true signals [138].

On the basis of an epigenome-wide association study in chronic headache by Winsvold et al. [44], Pereda et al. assessed the role of DNA methylation of the first exon of the *NPTX2* gene and the 5′ upstream region of the *SH2D5* gene in migraine chronification [45]. A sample of 109 chronic migraine patients, 98 EM patients, and 98 HCs was investigated. The *SH2D5* gene encodes the SH2 domain-containing protein 5, which regulates synaptic plasticity through the control of Rac-GTP levels. The *NPTX2* gene encodes the neuronal pentraxin II protein, an inhibitor of excitatory synapses, which acts by binding and clustering glutamatergic AMPA receptors [44]. No difference in DNA methylation levels was found among chronic migraine patients, EM patients, and HCs in the investigated regions of *NPTX2* and *SH2D5* genes, despite their role in the regulation of synaptic plasticity, which is one of the proposed mechanisms underlying the chronification of headache [33,139] (Figure 1).

A pilot study, which enrolled 25 patients with MOH, 20 EM patients, and 13 HCs aimed to identify changes in DNA methylation associated with headache chronification between selected groups. In all enrolled subjects, genome-wide DNA methylation analysis was performed. Although no statistical significance was found between groups, Terlizzi et al. identified some differently methylated CpG sites of interest, linked to the genes *COMT* (chr10:76993892 island), *GIT2* (chr12:110433797-110434205*Island), *ZNF234* (chr19:44645494-44646069*N_Shore), and *SOCS1* (chr16:11348541-11350803*Island). These findings imply that the mentioned genes may play a role in drug addiction and migraine progression to MOH, although a replication of the observed data in a larger sample is needed to further clarify and confirm their role in MOH development [46].

Epigenetic clocks have been analyzed in physiological and pathological conditions [140], and increased predicted epigenetic age compared with chronological age is associated with multiple conditions including neurological diseases [141,142]. It can be hypothesized that this mechanism may also play a role in migraine chronification. Kwiatkowska et al. [47] investigated associations between the epigenetic age and chronic pain, including a dataset of migraine patients, who were compared to HCs, previously studied by Terlizzi et al. [46], as mentioned above. However, the study revealed no significant difference in epigenetic age acceleration between MOH and HC cases or between EM and HC cases in the investigated dataset. Overall, the relationship between aging and chronic pain has so far been poorly investigated, with only one study that observed a younger epigenome in patients with chronic pain [143]. Additional studies in independent cohorts are required to better characterize chronic pain conditions, including chronic migraine, by epigenetic biomarkers of age.

## 7. Changed Histone Acetylation Patterns in Migraine

Neuroplastic changes play an important role in a variety of chronic neuropsychiatric conditions. In this context, epigenetic alterations through HDACs are frequently investigated. As was mentioned before, HDACs deacetylate histones, thus promoting chromatin condensation and altered gene expression [144,145]. However, some HDACs can also deacetylate non-histone targets, including proteins involved in cytoarchitecture and dynamic cellular structure. HDAC6 is primarily expressed in the cytosol, and one of its primary targets for deacetylation is α-tubulin [146,147]. To determine if altered neuronal cytoarchitecture facilitates the chronic migraine state and whether this state is reversible by inhibition of HDAC6, Bertels et al. [48] studied chronic migraine-associated pain in a nitroglycerin mouse model. Nitroglycerin-treated mice demonstrated decreased neuronal complexity in the somatosensory cortex, periaqueductal gray, and trigeminal caudal nucleus. The study further demonstrated that treatment with an HDAC6 inhibitor reversed these cytoarchitectural changes. These results may suggest a novel mechanism for migraine pathophysiology and establish HDAC6 as an innovative therapeutic target for migraine treatment (Figure 1). Protein kinases, such as c-Jun N-terminal kinases (JNKs) that belong to the mitogen-activated protein kinase family, relay, amplify, and integrate signals from a diverse range of intra- and extracellular stimuli. JNK pathways are activated in response to a wide range of stimuli, most notably following exposure of the cell to a variety of stress events [148]. C-Jun is an inducible transcription factor that activates the AP-1 DNA-binding complex. The binding of this complex to a specific DNA site close to a promotor or enhancer is crucial for the initiation of transcription [149,150]. Phosphorylation of c-Jun is required for nuclear translocation and the formation of the DNA-binding complex, which is catalyzed by JNKs [148,151]. Wu et al. [49] studied the role of the JNK/c-Jun cascade in the regulation of H3 acetylation in a rat TG model after stimulation by the neuro-inflammatory agent mustard oil. The results showed a significantly increased expression of phospho-JNK1 and phospho-c-Jun in TG neurons after mustard oil stimulation compared to neurons treated with a control agent (mineral oil). A significantly increased level of acetyl-H3 was also observed. Obtained results suggest a potential role of the JNK/c-Jun pathway in chromatin remodeling in TG neurons following stimulation. Such stress-induced histone modifications may be involved in migraine development. However, to the best of our knowledge, there are no studies to date that have investigated the role of JNK/c-Jun-induced H3 acetylation in TG in migraine pathogenesis.

Available data on the association of histone acetylation and migraine in mouse models are extremely limited. Acetylation modification is involved in many processes in brain tissue, such as glial cell proliferation, alterations in the activity of ion channels and neurotransmitter receptors, the plasticity of neurons, and the remodeling of neural networks. To date, many selective and multitargeting HDACis have been developed, with some of them being approved for the treatment of cancer, and some of them expected to become antiepileptic drugs [110,152]. Thus, further studies investigating the role of histone acetylation in episodic and chronic migraine are important as they may uncover novel putative drug targets for migraine treatment.

## 8. Epigenetics of Specific Pathways in Migraine

### 8.1. CGRP, RAMP1, and Migraine

CGRP system overview. A large number of studies have shown that the neuropeptide CGRP plays a key role in migraine pathogenesis [153,154]. CGRP is encoded by the *CALCA* gene, which codes for both CGRP and the hormone calcitonin as alternative splice products [155]. CGRP and its receptors are widely expressed in trigeminal neurons [156], where CGRP triggers neurogenic inflammation, by acting as a potent vasodilator through stimulating vascular smooth muscle adenylyl cyclase [157,158]. In migraineurs, intravenous CGRP administration induces migraine-like symptoms; during acute attacks, elevated CGRP blood levels are observed during active migraine episodes, as well as in chronic versus episodic migraineurs [159,160,161,162]. CGRP receptors, membrane heterodimer complexes, are composed of the calcitonin receptor-like receptor (CLR) and an accessory protein called RAMP1. The RAMPs, a small family of three proteins, are single transmembrane proteins that alter the pharmacology, functionality, and cell trafficking of receptors of CGRP family peptides. To form a functional receptor of the CGRP peptide, CLR has to form a complex with RAMP1 to create the receptor. Due to the complexity of this system of peptide receptors, their expression in the trigeminal system is not yet clear and their possible functional roles are yet to discover. CLR and RAMP1 mRNA and protein expression was detected in several relevant regions, such as the periaqueductal gray, area postrema, pontine raphe nucleus, spinal trigeminal nucleus, and spinal cord [158,163,164,165].

#### 8.1.1. *CALCA* Gene Epigenetics in Migraine

Park et al. investigated the epigenetic regulation of the *CALCA* gene in rat and human model cell lines and primary cultures of rat trigeminal ganglia glia, showing that the epigenetic regulation can greatly affect *CALCA* gene expression in those cell lines [25]. Rubino et al. compared the DNA methylation of two CpG-rich islands in the distal and proximal promoter regions of the *CALCA* gene (−2762 to −2362 bp and −1662 to −1028 bp upstream, respectively, counted from ATG) in 22 patients (15 females) with episodic migraine (EM) without aura and 20 HCs (12 females). No significant differences in methylation were observed in the distal promoter region in migraineurs compared to controls. However, the investigation of the proximal promoter region revealed a lower methylation level at two CpG sites (−1461 and −1415 bp). In addition, Rubino et al. analyzed the association of DNA methylation levels and clinical characteristics of migraine and observed that the methylation at CpG site −1461 is positively correlated with the age of onset of migraine, while the methylation at CpG site −1393 is inversely correlated with the presence of nausea or vomiting during migraine attacks [50] (Figure 1). The study provided the first evidence that the methylation of *CALCA* is reduced in patients with migraine, and that methylation may be associated with disease characteristics.

#### 8.1.2. RAMP1 Epigenetics in Migraine

Emerging evidence indicates that the altered expression of *RAMP1* can affect the sensitivity of cells to CGRP [164]. Nestin/RAMP1 transgenic mice, for example, that over-express human *RAMP1* in the central nervous system show some migraine-typical features, such as photophobia and allodynia, after CGRP administration [166,167]. First attempts to analyze DNA methylation at the promoter region of *RAMP1* in 26 migraineurs compared to 25 HC was performed by Wan et al. in 2015 [51]. The study revealed no significant difference in DNA methylation level among 13 detected CpG sites or units at the *RAMP1* promoter region between migraine and control groups. Further stratification showed that the methylation levels at CpGs +25, +27, and +31, related to TSS, were significantly higher in migraineurs with a migraine family history compared to those without it. In addition, methylation level at the CpGs +89, +94, and +96 was significantly lower in female migraineurs compared to that in female HCs. A subsequent stratification according to the abundance of photophobia in migraineurs did not reveal any methylation differences. The obtained results may suggest a role of CpG methylation in female migraine risk. Carvalho et al. investigated the *RAMP1* promoter methylation in 54 female migraineurs against 50 controls and detected that the CpG site −284 in the *RAMP* promoter showed significantly higher methylation rates in migraineurs [24]. However, it is worth noting that Carvalho et al. obtained higher promoter methylation levels in female migraineurs compared to female controls, which contradicts the results of the previously mentioned study [24].

These data suggest that *CALCA* and *RAMP1* epigenetic regulation may play a considerable role in migraine pathogenesis. However, it becomes clear that only a limited number of studies with a small sample size have been performed, highlighting the urgent need for further investigations based on robust samples to validate the presented findings.

### 8.2. The Endocannabinoid System and Migraine

#### 8.2.1. The Endocannabinoid System—A Short Overview

The endocannabinoid system (ES) has an important role in the regulation of neuronal function, particularly in the regulation of synaptic function and neurodevelopment. The endogenous ligand (eCB) signaling system consists of (1) at least two G-protein-coupled receptors, known as the cannabinoid type-1 and type-2 receptors (CB1R and CB2R), (2) eCBs, of which anandamide (AEA) and 2-arachidonoylglycerol (2-AG) are the best characterized, and (3) synthetic and degradative enzymes and transporters that regulate eCB levels and action at receptors [168].

#### 8.2.2. General Aspects of the Epigenetic Regulation of the Endocannabinoid System

Numerous studies have reported that the ES undergoes epigenetic modulation by alcohol, diet, stress, smoking, exercise, or drugs. Such epigenetic changes may alter the expression of ES components and, subsequently, ES signaling. The principal targets of epigenetic changes are the genes responsible for encoding cannabinoid receptors, particularly *CNR1*, encoding CB1, and *FAAH*, encoding fatty acid amide hydrolase (FAAH), a hydrolyzing enzyme, which leads to subsequent alterations of ES tone. The detected epigenetic mechanisms involve changes in DNA methylation (global and gene-specific), histone tail modifications such as acetylation, deacetylation, or methylation, and the production of specific miRNAs in different brain regions, peripheral tissues, and cell lines [169]. Moreover, it has been reported that stress induces epigenetic changes in the ES. Lomazzo et al. [170] found a reduction in the levels of histone H3K9 acetylation (H3K9ac) associated with the *Cnr1* gene in mice. Chronic stress has also been associated with increased methylation of the *Cnr1* gene promoter by DNMT1, resulting in reduced levels of CB1 in the sensory neurons that innervate pelvic viscera in mice. Furthermore, chronic stress increases the expression of the histone acetyltransferase EP300 and promotes the acetylation of histones in the *Trpv1* promoter (the gene encoding for AEA) in mice, thus increasing levels of TRPV1 in these neurons. These observations indicate that chronic stress promotes DNA methylation and downregulation of antinociceptive Cnr1 and a concurrent increase in histone acetylation of pronociceptive Trpv1, resulting in visceral hyperalgesia [171]. However, the role of stress-driven epigenetic changes of the ES in brain disorders such as migraine is still unsettled and requires further research.

The epigenetic regulation of the ES represents an important research topic, considering that ES components are found dysregulated in different pathological conditions such as obesity, diabetes, colorectal cancer, schizophrenia, and Alzheimer’s, Parkinson’s, and Huntington’s diseases [172].

#### 8.2.3. Epigenetics of the Endocannabinoid System in Migraine

Epigenetic changes in the ES have been detected in several pathological situations such as glioblastoma, colorectal cancer, and Alzheimer’s disease [169]. Reduced DNA methylation at the *FAAH* gene promoter (responsible for encoding FAAH) has been reported in late-onset Alzheimer’s disease [173]. Notably, patients with the most severe cognitive impairment exhibited the lowest levels of methylation, suggesting FAAH as a potential therapeutic target for Alzheimer’s disease. Such an association may imply an important role of epigenetic modifications of ES in brain function, thus suggesting their role in other neurological disorders such as migraine. ES modulates the function and activity of signaling pathways that are involved in pain control and plays a crucial role in migraine pathogenesis [174,175,176]. ES has been shown to be altered in experimental models of different neurological disorders and in plasma and postmortem brain samples from humans with these disorders [177]. Greco et al. compared the expression and DNA methylation levels of genes of ES components among 25 EM patients, 26 chronic migraine patients with MOH, and 24 HCs. Despite the observation of significantly different expression levels of the genes *CB1*, *CB2*, *FAAH*, *NAPE-PLD*, *MAGL*, and *DAGL* in migraineurs compared to HCs, DNA methylation analysis did not show any significant differences between patients with migraine and HCs at the detected CpG sites at the promoter region levels of all the evaluated genes [52]. It is possible that other epigenetic mechanisms are implicated in gene expression regulation of the ES in migraine, such as DNA hypomethylation and histone hyperacetylation, which have been shown to regulate *CB1* and *CB2* gene expression in the cells of the immune and nervous systems. Thus, further studies of epigenetic regulations of ES are required to unravel this topic [178].

#### 8.2.4. The Endocannabinoid System Influences Epigenetics

Although the ES is regulated by epigenetic mechanisms, the ES itself is also able to induce epigenetic alterations through phytocannabinoids, endocannabinoids, and endocannabinoid receptor agonists/antagonists. The ES can promote epigenetic changes by regulating, e.g., DNA methylation or histone modifications that, in turn, can induce changes in the expression of genes that play a key role in neurodevelopment. Moreover, CB1R activation has been reported to promote changes in the expression of genes that play an important role in various neurotransmitter systems [179]. AEA has been reported to protect neurons from inflammatory damage by inducing histone H3 phosphorylation of Mpk-1 (the gene encoding for MAPK phosphatase-1) in activated microglial cells, thus regulating Mpk-1 expression and subsequently dephosphorylating ERK1/2 [180]. Furthermore, it has been documented that alcohol exposure induces DNA methylation changes in the mouse model of fetal alcohol spectrum disorder, and the lack of a functional *CNR1* gene protects against ethanol-induced impairments of DNMT1, DNMT3A, and DNA methylation [181].

Exogenous Δ9-tetrahydrocannabinol (THC) has been demonstrated to cause epigenome alteration in rats [172]. Exposure of adolescent male rats to THC has been shown to affect the transcriptional and epigenetic state of penk (the gene encoding for opioid neuropeptide proenkephalin) through repression of histone H3K9 methylation in the adult nucleus accumbens, [182]. Prini et al. showed significantly increased H3K9me3 levels in the prefrontal cortex of THC-exposed animals compared to controls [183]. The authors also observed that a THC-mediated increase in H3K9me3 levels promoted chromatin changes in a set of genes whose expression was downregulated following THC exposure, namely, Homer1, Mgll, and Dlg4. Gerra et al. showed that cannabis users presented hypermethylation of the exon 8 of *DRD2* (the gene encoding for dopamine receptor D2), as well as the CpG-rich region of the *NCAM1* (the gene encoding CD56), compared to controls [184].

The complex interplay between ES and epigenetics and the ability of ES to affect epigenetic modifications of specific genes may represent a potential epigenetic target for the treatment of diseases including migraine, as well as the development of possible epigenetic therapies. Nevertheless, the role of epigenetic changes in ES in migraine has scarcely been researched, and further detailed investigation of ES epigenetics on bigger samples of patients with episodic and chronic migraine is required for clarification.

### 8.3. The Epigenetic Regulation of Other Pathways in Migraine

Some of the genetic factors linked to migraine have also been linked to epigenetic mechanisms. Examples are functionally relevant polymorphisms known for the gene expressing *MTHFR*, which is involved in the pathway for generating the methyl donor required for DNA methylation. These variants are associated with migraine according to some studies [185]. In addition to *MTHFR*, other genes, such as *MTDH*, *MEF2D*, and *PRDM16*, also affect epigenetic processes and have been linked to migraine pathophysiology. Multiple genome-wide association studies have identified polymorphisms associated with migraine. Most recently, Hautakangas et al. identified 123 risk loci for migraine [186]. However, the role of epigenetic modifications affecting these polymorphically expressed risk genes remains unclear and needs to be further studied.

## 9. Epigenetics as a Therapeutic Target in Migraine

On the basis of the initial evidence that epigenetics plays a role in migraine, it can be assumed that epigenetics may also be valuable as a therapeutic target in migraine. A number of epigenetic therapeutics are approved for the treatment of various cancers, such as myelodysplastic syndrome, certain types of leukemia, large B-cell lymphoma, and cutaneous T-cell lymphomas, with even more being currently developed for cancer treatment. Of the nine epigenetic therapeutics approved by the US Food and Drugs Administration, two are DNMT inhibitors, four are the aforementioned HDACs, two are isocitrate dehydrogenase inhibitors (IDHs), and one is the EZH2 inhibitor tazemetostat. Generally, anticancer drugs have been developed that target DNMTs, HATs, HDACs, histone demethylases, HMTs, and IDHs. IDHs inhibit TET enzymes. Thus, IDH inhibitors relieve TET enzyme inhibition, leading to anticancer effects [187]. However, in the case of migraine, the relationship between specific epigenetic targets and the pathophysiology of migraine needs to be further established to develop novel epigenetic therapeutic agents.

## 10. General Aspects of microRNAs and circRNAs in Migraine

One of the most relevant and promising areas of epigenetics is the field of microRNA- and circRNA-dependent regulation of expression and its influence on physiological and pathological processes in cells under various conditions, including migraine. Migraine is a multifactorial disease, and a variety of exogenous and endogenous triggers of migraine development and attacks may be at least partly mediated by changes in circRNA-dependent regulation of expression [188].

Key aspects of migraine pathogenesis appear to be complex disturbances in the interaction between nociceptive and antinociceptive systems, as well as processes such as neuroinflammation, neurovascular conflict, and central sensitization, which lead to the characteristic and complex appearance of headache and additional accompanying neurological symptoms [189]. These pathological conditions are based on the dysfunction of neurotransmitter systems, their receptors, and intracellular signaling pathways affecting inflammatory response, which are assumed to be largely controlled by microRNAs [190].

## 11. Principal Mechanisms of Action of microRNAs and circRNAs

### 11.1. microRNAs—Key Functions and Mechanisms of Action

microRNAs are small molecules consisting of 20–25 nucleotides that do not have their own coding properties [191]. Their key function is to regulate expression epigenetically by suppressing or enhancing transcription and/or translation of matrix RNAs [191]. Two main mechanisms of microRNA functioning are distinguished [192]. The first one is related to direct binding to mRNA and its subsequent degradation. The second mechanism leads to a reduction of protein synthesis by ribosome inhibition and, thus, inhibition of mRNA translation [192]. In addition to the described mechanisms, of note, microRNAs can affect the methylation of regulatory DNA sites by changing the activity of DNA methyltransferases [193,194]. microRNAs either act as unbound, free microRNA molecules or through the formation of the RNA-inducible gene turn-off complex, consisting of microRNAs and proteins of the Dicer and Ago families [195].

### 11.2. circRNAs: Key Functions and Mechanisms of Action

circRNAs are a recently discovered large class of RNAs whose key property is a ring structure in which the 3′ and 5′ ends are covalently linked [196]. So far, it has been discovered that some circRNAs can encode proteins, but the regulatory functions of circRNAs are the most studied property [196,197]. The mechanisms via which circRNAs are involved in epigenetic regulation are diverse [198,199]. circRNAs can act as a “sponge” for miRNA, thus influencing miRNA-dependent epigenetic regulation [200]. Moreover, circRNAs affect pre-mRNA splicing and can bind to RNA-binding proteins, affecting mRNA stability [201,202]. Lastly, some circRNAs can undergo translation, which confirms the coding properties of some members of this RNA class [203].

## 12. Changed Expression Patterns of microRNAs and circRNAs in Migraine

In recent years, initial studies on the role of microRNAs in the pathophysiology of migraine were published. Those studies showed that certain microRNAs are closely related to important migraine-related pathomechanisms, such as neuroinflammation, neurovascular conflict, vasoconstriction, and central sensitization, as well as to the clinical manifestations of migraine, including pain severity and the presence of aura.

### 12.1. The Role of microRNA in the Pathophysiology of Migraine

#### 12.1.1. The Role of microRNA in the Pathophysiology of Migraine in Animal Models

Hsa-microRNA-155-5p. A possible approach is to study the role of microRNAs in migraine using animal models. However, we only found one study on this topic. Using a nitroglycerin-induced mouse chronic migraine model, Wen et al. demonstrated that microRNA-155-5p may play a role in migraine pathogenesis by inhibiting SIRT1, which is involved in neuroinflammatory processes and central sensitization [53]. Further research on this issue in humans is needed to clarify the relevance of microRNA-155-5p for clinical practice.

#### 12.1.2. The Role of microRNA in the Pathophysiology of Migraine in Humans

Hsa-miR-5189-3p, hsa-miR-96-5p, hsa-miR-3613-5p, hsa-miR-99a-3p, and hsa-miR-542-3p. In their study, Timea Aczél et al. examined microRNA expression profiles in peripheral blood mononuclear cells in 12 healthy controls and 16 migraineurs between and during pain attacks [54]. There were statistically significant differences in the expression profiles of 31 microRNAs between the groups, including hsa-miR-5189-3p, hsa-miR-96-5p, hsa-miR-3613-5p, hsa-miR-99a-3p, hsa-miR-542-3p. Differences in microRNA expression profiles in patients between attacks and during migraine attacks were also observed. On the basis of the analysis of altered molecular pathways, the authors hypothesized a correlation between microRNA expression levels in immune cells and the activity of signaling pathways involving Toll-like receptors, cytokine receptors, neuroinflammation, and oxidative stress, which are significant for migraine pathogenesis.

Has-miR-34a-5p and has-miR-382-5p. Andersen et al. studied the microRNA expression profiles in 28 migraineurs between and during headache attacks and in a control group of 20 healthy subjects [34]. The authors showed that migraine attacks are associated with an upregulation of miR-34a-5p and miR-382-5p expression. The in silico predicted target genes for these microRNAs were *GABBR2, SLC6A1*, and *GABRA3* for miR-34a-5p and *IL10RA* and *GABRA5* for miR-382-5p, respectively [34].

Circadian gene *CLOCK*, financial stress, and microRNAs. Baksa et al. demonstrated that financial stress may be a causal factor for migraine through altered expression of the circadian *CLOCK* gene [55]. The *CLOCK* gene is one of the central regulators of circadian rhythms in the cell and controls many metabolic processes [204]. Mutations and polymorphisms of the *CLOCK* gene may be associated with mood disorders, sleep disturbances, and cancer [204]. In a cohort of 2157 subjects, the authors examined the relationship of the *rs10462028* polymorphism of the *CLOCK* gene with three key stressors (childhood adversity, recent negative life events, and financial difficulties) and the likelihood of developing migraine based on questionnaires and genetic testing. The authors suggest that financial stress may alter *CLOCK* expression levels at the epigenetic level, including changes in microRNAs, probably through alteration of miRNA binding in the 3′UTR of the *CLOCK* gene.

The role of microRNAs in endothelial dysfunction. Chen et al. examined a panel of microRNAs associated with migraine and reversible cerebral vasoconstriction syndrome, including 30 patients during the ictal stage and 30 patients during the interictal stage, as well as 30 age- and sex-matched controls [56]. It was demonstrated that the expression of let-7a-5p, let-7b-5p, and let-7f-5p was enhanced in patients during the ictal stage compared to the interictal stage and the control group. At the same time, the authors showed that higher miR-130a-3p expression was associated with impaired blood–brain barrier permeability in patients showing reversible cerebral vasoconstriction. In a separate study, the same authors observed an increased expression of miR-155, miR-126, and let-7g in 30 migraineurs compared to the 30 healthy controls, with expression levels of these microRNAs positively correlated with ICAM-1 levels, a marker of endothelial dysfunction [57]. Both studies are important for understanding the functionally relevant role of these microRNAs in the vascular component of migraine pathogenesis.

### 12.2. microRNA and Treatment of Migraine

microRNAs and the anti-CGRP agent erenumab. Some studies have investigated drug-induced changes in microRNA expression in migraineurs, which are of particular interest. De Icco Roberto et al. showed that miR-382-5p and miR-34a-5p expression significantly decreased after 84 days of first erenumab administration in a group of 40 migraineurs. However, no statistically significant relationship between the response to treatment and the expression levels of these microRNAs was found [58] (Figure 2).

microRNA and NSAIDs. A study by Galleli et al. included 24 migraineurs (12 treated with acetaminophen or a combination of ibuprofen and magnesium and 12 untreated) and observed a 50% decrease in hsa-miR-34a-5p and hsa-miR-375 expression in the group of treated patients compared to untreated patients regardless of sex and age [59] (Figure 2).

microRNAs as prognostic markers for migraine. Interestingly, of note, the work of Greco et al. demonstrated that chronic migraine patients with medication overuse showed increased plasma CGRP levels and an increased expression of miR34a-5p and miR-382-5p, as measured in peripheral blood mononuclear cells compared to individuals with EM [26]. Altogether the mentioned studies suggest a putative potential of miR-382-5p and miR-34a-5p as possible diagnostic and prognostic markers of migraine severity and response to therapy. However, more studies are needed to prove the first collected results.

### 12.3. microRNAs and Clinical Characteristics of Migraine

microRNAs and migraine without aura. In a study by Tafuri et al., which included 15 migraineurs without aura and 13 healthy controls, the authors showed that miR-27b expression was elevated while miR-181a, let-7b, and miR-22 expression levels were decreased in the group of migraineurs without aura compared with healthy controls [60] (Figure 2). Studies investigating the association of changes in the microRNA profile and migraine subtypes are still very limited but needed, as they would help to better understand whether those changes would be useful for a more accurate differentiation between clinical types of migraine.

microRNAs as a marker for migraine severity. Zhai et al. could demonstrate a direct involvement of miR-30a in migraine pathogenesis [61]. They showed that miR-30a levels were significantly reduced, and that miR-30a methylation in the promoter region was enhanced in migraineurs compared to the control group. Through bioinformatic analysis, the authors showed that *CALCA* encoding CGRP is a target of miR-30a. Using Western blot analysis, the authors further demonstrated that miR-30a knockdown leads to increased *CALCA* expression; conversely, increased miR-30a expression suppresses *CALCA* (Figure 2). Most importantly, miR-30a levels were significantly reduced in the group of patients with bilateral seizures, persistent pain, and a high pain index, suggesting a putative potential of miR-30a as a marker of severity and prognostic marker in patients with migraine.

### 12.4. The Role of circRNAs in Migraine Pathophysiology

Another possible epigenetic marker significant for migraine could be circRNA. Lin et al. identified 794 and 1245 circRNAs, which were increased and decreased, respectively, in migraineurs compared to healthy controls [62]. The authors concluded that hsa_circRNA_103670, hsa_circRNA_103809, hsa_circRNA_000367, and hsa_circRNA_102413 may be closely related to the development of migraine. However, because of the small sample size in this study and the limited number of studies that have hitherto investigated circRNAs in relation to migraine, it is difficult to draw firm conclusions on the clinical relevance of circRNAs in migraine.

## 13. Discussion

The interest in the role of epigenetics in migraine development is growing, with an increasing number of studies being published in the field. We explored the current knowledge gathered in experimental studies, as well as animal and human in vivo studies, regarding epigenetics changes and their role in migraine development, severity, and migraine treatment. Focus was specifically laid on methylation, histone acetylation, and microRNA shifts in association with migraine.

This review identified *DGKG*, *CALCA*, *RAMP1*, *SH2D5*, *NPTX2*, *COMT*, *GIT2*, *ZNF234*, and *SOCS1* as genes considerably affected by methylation shifts in migraine. Studies devoted to methylation have especially focused on the epigenetic changes of the CGRP system. It has been demonstrated in animal models and humans that *CALCA* expression is highly dependent on epigenetic regulation, especially the level of CpG methylation. It is particularly interesting that methylation of different CpG sites in the *CALCA* gene is associated with various clinically relevant migraine characteristics, such as the age of migraine onset and the presence and severity of nausea and vomiting during migraine attacks. It should be noted that studies on the epigenetics of *RAMP1*, one of the important members of the CGRP system, are largely contradictory and very limited. From the currently available literature, we can conclude that more extensive studies devoted to the epigenetics of *CALCA* and *RAMP1* are needed for a more detailed analysis of the correlations between methylation patterns of these genes and the key mechanisms of migraine pathogenesis, as well as important clinical characteristics, prognostic factors, and perhaps response to therapy in migraineurs. A candidate gene for which methylation is closely related to the pathophysiology of migraine is *DGKG*, but there are currently insufficient data to assess the clinical relevance of epigenetic changes in this gene. Another important aspect of migraine epigenetics is the study of the significance of epigenetic factors in the transition of episodic migraine to the chronic form, as well as its role in the development of dependence and resistance to analgesics in migraineurs. Currently, *SH2D5* and *NPTX2* can be recognized as the most interesting genes in this respect. However, previous publications devoted to this topic are contradictory in their results. One study showed that methylation of some CpG sites of *COMT*, *GIT2*, *ZNF234*, and *SOCS1* genes could be important for drug addiction and transformation of migraine into MOH, but studies on larger patient cohorts are needed to fully explore this issue. One of the most interesting and understudied areas remains the study of epigenetic changes in the endocannabinoid system, especially the regulation of *CB1* and *CB2* genes, as well as their significance for understanding the pathophysiology and improving migraine therapy.

There have been very interesting studies investigating microRNA molecules in relation to migraine. MiR-34a-5p, miR-382-5p, let-7a-5p, let-7b-5p, let-7f-5p, miR-155, miR-126, let-7g, hsa-miR-34a-5p, hsa-miR-375, miR-181a, let-7b, miR-22, and miR-155-5p appear to play a considerable role in migraine pathogenesis and therapy response in migraine. The setup of studies on the relationship between microRNAs and migraine is not fundamentally different from that of other aspects of migraine epigenetics mentioned above. Thus, these studies also showed both limitations and drawbacks, such as small and unrepresentative sample sizes and some difficulty in interpreting changes in gene expression in peripheral blood cells and their correlation with molecular shifts in the brain.

The specificity of microRNA studies lies in the difficulty to identify the target gene and fully assess the range of functions of a particular microRNA in the molecular biology of the cell and pathophysiology of a disease. Currently, we can identify several key groups of microRNAs involved in the development of headache attacks, responsible for treatment response and chronification of migraine. The first group includes miR-34a-5p, miR-382-5p, let-7a-5p, let-7b-5p, let-7f-5p, miR-155, miR-126, let-7g, and miR-30a. MiR-34a-5p and miR-382-5p may potentially be associated with response to erenumab therapy. In addition to identified correlations of miR34a-5p and miR-382-5p expression levels with the occurrence of migraine attacks and response to therapy, it has also been demonstrated that these microRNAs may be associated with the development of medication overuse. Of particular importance for clinical practice is the study of markers such as miR-27b, miR-181a, let-7b, and miR-22, which are associated with the presence of aura in migraineurs. Those molecules may have the potential to become useful tools for diagnosis, assessment of migraine severity, and potential targets for migraine therapy. However, more research on this topic in larger studies is needed.

Important drawbacks of the current studies of migraine epigenetics relate to the low power of these studies and the difficulty of interpreting the results. Most human studies are based on the assessment of peripheral blood parameters in patients, as a proxy for the real biochemical changes in the brain. Another uncertain and limiting aspect is the lack of animal models that are adequately representative of the disease. For example, Labruijere et al. demonstrated that data collected from rat peripheral blood leukocytes were unrepresentative of changes in methylation patterns in the trigeminal caudal nucleus and TG. Unfortunately, only a few small studies are currently available on animal models dealing with histone acetylation patterns in migraine. The significance of histone acetylation in the pathophysiology of migraine in patient cohorts is not yet sufficiently studied. The study of JNK/c-Jun cascade as a future target for migraine therapy [49], as well as the study of HDAC6 inhibitors [48] as potential drugs for migraine treatment, can be considered very promising, which underlines the need for more studies regarding the role of histone acetylation in migraine.

With the growth in knowledge of the role of epigenetics in migraine pathophysiology and treatment, the field of headache research will soon also be confronted with putative ethical implications that a rapidly developing field such as epigenetics has to deal with in general. The possibility of epigenetic engineering, which involves the intentional alteration of the epigenome to achieve desired outcomes, such as disease prevention or the augmentation of certain disease traits, needs to be carefully balanced and evaluated regarding the risk/benefit and social and economic implications, in association with the accessibility of the technology. Other ethical issues of epigenetics comprise the possibility to prevent certain epigenetic triggers of a disease due to a preventive parental behavior, e.g., through the adaptation of certain lifestyle factors that may also be relevant topics for migraine in the future when the disorder is better epigenetically understood [205,206,207].

## 14. Conclusions

We conclude that the study of migraine epigenetics is still at a very early stage, but promising data are being produced at an increasing speed. At the time of our review, we were able to find very few confirmative studies on this topic. Clinical studies on migraine epigenetics are characterized by small numbers of participants and inconsistent methodology. Furthermore, results from studies in animal models are difficult to interpret in the context of migraine in humans. These limitations are particularly characteristic for studies on the relationship between epigenetics and the clinical features of the course, severity, and response to therapy of various forms of migraine in different groups of patients, which are important for clinical practice in the era of personalized medicine. The most promising targets for study involve the methylation of CGRP system genes, such as *CALCA* and *RAMP1*. A larger study of epigenetic changes in *SH2D5*, *NPTX2*, *COMT*, *GIT2*, *ZNF234*, and *SOCS1* genes is necessary to understand the processes of migraine chronicity and MOH development. Studies of the endocannabinoid system, as well as histone acetylation processes and the JNK/c-Jun cascade as potential targets for migraine therapy, seem very promising. Studies of microRNAs, especially miR-34a-5p and miR-382-5p, have the potential to shed light on questions of migraine pathophysiology, response to therapy, and chronicity of headache. Larger studies, over longer running times, are needed to confirm the hitherto gathered results.

## Figures and Tables

**Figure 1 ijms-24-09127-f001:**
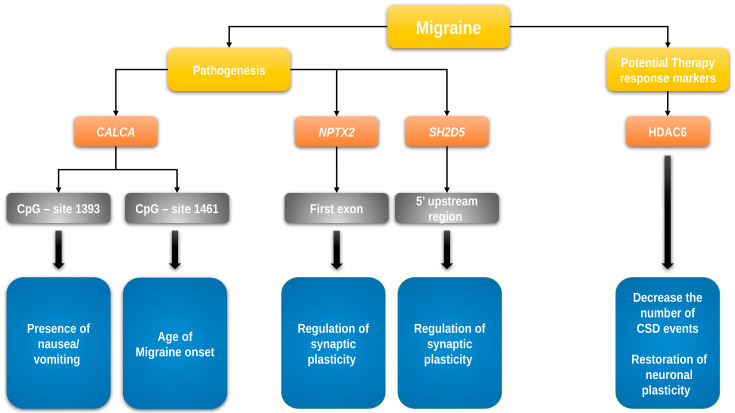
DNA methylation/demethylation and histone acetylation/deacetylation in migraine. Yellow: general categories; orange: gene name; gray: affected gene site; blue: function and role in migraine pathogenesis/clinical characteristics. The figure shows the genes whose epigenetic changes through methylation/demethylation and/or acetylation/deacetylation of histones may play a significant role in the pathophysiology and clinical course of migraine. CALCA—gene encoding calcitonin-related polypeptide alpha, HDAC6—gene encoding histone deacetylase 6, NPTX2—gene encoding neuronal pentraxin-2, and SH2D5—gene encoding SH2 domain-containing 5 protein.

**Figure 2 ijms-24-09127-f002:**
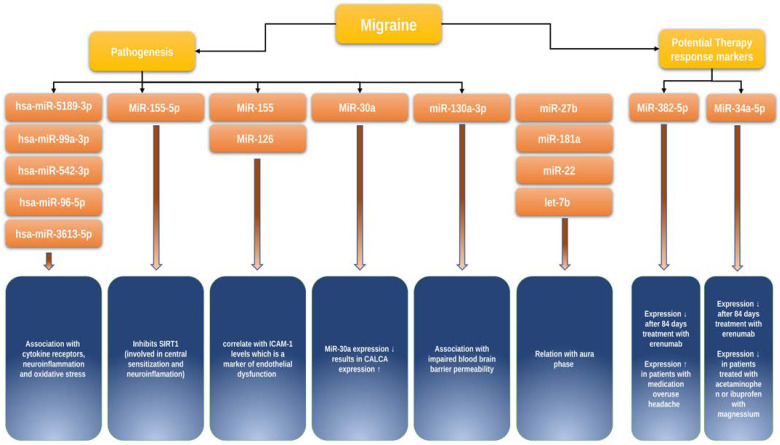
MicroRNAs in migraine. Yellow: general categories; orange: microRNA name; blue: function and role in migraine pathogenesis/clinical characteristics. The figure highlights key microRNA molecules currently known to be important in migraine, i.e., their functions and role in migraine development, clinical characteristics, and response to therapy. SIRT1—gene encoding Sirtuin 1, ICAM-1—intercellular adhesion molecule 1.

**Table 1 ijms-24-09127-t001:** Studies investigating the role of histone modifications in migraine development.

Study	Title	First Author	Sample	Ethnicity	Genes/Molecules Assessed	Findings
[24]	A high methylation level of a novel −284 bp CpG island in the RAMP1 gene promoter is potentially associated with migraine in women	Carvalho Estefânia	104 females (54 migraineurs, 50 controls)	Portuguese	*RAMP1* gene promotor methylation	5 differently methylated CpG dinucleotides (−346, −334, −284, −276, and −234 ^1^) were assessed; −284 CpG unit found to be significantly methylated in migraineurs
[25]	Epigenetic regulation of the calcitonin gene-related peptide gene in trigeminal glia	Ki-Youb Park	Rat and human model cell lines; primary cultures of rat TG ^11^ glia	-	DNA methylation and histone acetylation in the promotor region of the *CALCA* gene	CT and CGRP mRNAs were assessed; CpG island methylation and histone H3 acetylation at the 18 bp cell-specific enhancer correlated with *CALCA* gene expression
[41]	Methylation of migraine-related genes in different tissues of the rat	Labruijere, S.	Rat tissues (leukocytes, thoracic aorta, dura mater, TG, caudal nuclei); 395 healthy women	-	Comparison of DNA methylation of migraine-specific genes in different migraine-related tissues: *Calca*, *Ramp1*, *Crcp*, *Calcrl*, *Usf2*, *Esr1*, *Gper*, *Nos3*, *Mthfr* Comparison of DNA methylation of migraine-specific genes between rats and humans	Methylation of the *Crcp*, *Calcrl*, *Esr1*, and *Nos3* genes is tissue-specific; methylation in leukocytes does not correlate with that in other tissues
[42]	Using monozygotic twins to dissect common genes in posttraumatic stress disorder and migraine	Charlotte K Bainomugisa	Total: 42 monozygote twins PTSD: 12 participants (10 males, 2 females) Migraineurs: 30 participants (14 males, 16 females)	Caucasian	Genome-wide DNA methylation levels assessment	Differently methylated genes *ADCYAP1*, *AIM2*, *CRHR1*, *DBH*, *DOCK2*, *FKBP5*, *HTR3A*, *OXTR*, *RORA*, *WWC1*, and *TSNARE1*
[43]	Genome-wide DNA methylation profiling in whole blood reveals epigenetic signatures associated with migraine	Gerring Zachary F.	67 migraineurs and 67 age- and sex = matched controls	Northern Europeans	Epigenome-wide analysis of differently methylated regions	No single methylation probe reached genome-wide significance
[44]	Epigenetic DNA methylation changes associated with headache chronification: A retrospective case–control study	Winsvold, B.S.	36 female headache patients 35 controls with episodic headache	Norwegian	Assessment of DNA methylation at 485,000 CpG sites	Possible association of 2 CpG sites in *SH2D5* and *NPTX2* genes with chronification of episodic headache
[45]	Methylation analysis of NPTX2 and SH2D5 genes in chronic migraine: A case–control study	Sara Perez Pereda	109 CM patients; 98 EM patients; 98 controls	N/A	Assessment of methylation of two CpG sites related to NPTX2 ^7^ and SH2D5 ^8^	No significant differences in methylation levels between CM, EM, and HC ^9^ in the first exon of the *NPTX2* gene or the 50 upstream region of the *SH2D5* gene
[46]	Epigenetic DNA methylation changes in episodic and chronic migraine	Terlizzi, R.	18 MOH; 20 EM patients; 11 HC	N/A	Genome-wide DNA methylation levels association with headache chronification	Hypermethylation of chr10:76993892 island in *COMT* gene in MOH cases compared to HCs; hypomethylation of GpG site at chr12:110433797-110434205*Island in MOH cases compared to HCs (*GIT2* gene), at chr19:44645494-44646069*N_Shore in MOH cases compared to HCs (ZNF234 gene), and at chr16:11348541-11350803*Island in MOH cases compared to HCs (*SOCS1* gene)
[47]	Analysis of epigenetic age predictors in pain-related conditions	K. M. Kwiatkowska	22 MOH ^10^ patients; 18 EM patients; 13 HC	Italian	Association between epigenetic age and chronic pain, by investigating first- and second-generation epigenetic clocks and DNA methylation surrogates of plasma proteins, blood cell counts, and telomere length in headache conditions	No significant difference in epigenetic age acceleration, DNA methylation surrogates comprised in GrimAg, and estimates of telomere length and blood cell counts between MOH cases and HCs or between EM cases and HCs.
[48]	Neuronal complexity is attenuated in preclinical models of migraine and restored by HDAC6 inhibition	Bertels Z.	Mouse models of migraine	-	HDAC6 inhibition and its effect in mouse models of migraine	HDAC6 inhibition restored neuronal plasticity and decreased the number of cortical spreading depression events
[49]	JNK1 regulates histone acetylation in trigeminal neurons following chemical stimulation	Wu, J.	TG neuron culture	-	The role of JNK/c-Jun cascade in the regulation of acetylation of H3 following chemical stimulation in TG neurons	Mustard oil stimulation activated the JNK/c-Jun pathway significantly by enhancing phospho-JNK1, phospho-c-Jun expression, and c-Jun activity, which were correlated with elevated acetylated H3 histone in TG neurons
[50]	Analysis of the DNA methylation pattern of the promoter region of calcitonin gene-related peptide 1 gene in patients with EM: An exploratory case–control study	Elisa Rubino	22 EM patients; 20 controls	N/A	Evaluation of DNA methylation of *CALCA* gene in patients with EM	No differences in methylation of the 30 CpG sites at the distal region of *CALCA* were found in migraineurs compared with controls; no overall difference was found in the methylation level among these six detected CpG sites at the *CALCA* proximal promoter between migraineurs and controls; however, the DNA methylation profile in two CpG sites at the proximal promoter region of *CALCA* was lower in migraineurs when compared to HCs
[51]	DNA methylation of RAMP1 gene in migraine: An exploratory analysis	Dongjun Wan	26 migraineurs, 25 matched controls	Chinese	DNA methylation levels at *RAMP1* promoter region	No significant differences in 13 detected CpG sites or units at the *RAMP1* promoter region in migraineurs compared with HC
[52]	Peripheral changes of endocannabinoid system components in episodic and chronic migraine patients: A pilot study	Rosaria Greco	25 EM ^2^ patients (24 females) 26 CM ^3^-MO ^4^ (22 females) 24 Controls (18 females)	N/A ^5^	DNA methylation changes in genes involved in ES ^6^ components	Methylation of *CNR1*, *CNR2*, *DAGLA*, *FAAH*, *MGLL*, *NAPEPLD*, and *RPL19P12* was assessed; DNA methylation analysis did not show any significant differences between patients and controls with regard to the detected CpG sites at the promoter region levels of all the evaluated genes

^1^ Related to the TSS (transcription start site; ^2^ episodic migraine; ^3^ chronic migraine; ^4^ medication overuse; ^5^ not available; ^6^ endocannabinoid system; ^7^ neuronal pentraxin II protein; ^8^ SH2 domain-containing 5 protein; ^9^ healthy controls; ^10^ medication overuse headache; ^11^ trigeminal ganglion.

**Table 2 ijms-24-09127-t002:** Studies investigating the role of miRNAs in migraine development.

Study	Title	First Author	Sample	Ethnicity	Genes/Molecules Assessed	Molecules That Are Expressed Differently
[26]	Plasma levels of CGRP and expression of specific microRNAs in blood cells of episodic and chronic migraine subjects: Toward the identification of a panel of peripheral biomarkers of migraine?	Rosaria Greco	27 EM patients, 28 patients with CM-MO ^6^	N/A	Evaluation of plasma levels of CGRP ^7^ and the expression of miR34a-5p and miR-382-5p in peripheral blood mononuclear cells	CGRP, miR-382-5p, and miR-34a-5p levels were significantly higher in CM-MO subjects when compared to EM patients
[34]	Serum MicroRNA signatures in migraineurs during attacks and in pain-free periods	Andersen, H. H.	2 cohorts, 28 migraineurs, 20 HC ^3^	N/A	Serum microRNA profiles of migraineurs during attacks and pain-free periods compared with healthy controls	miR-34a-5p miR-29c-5p, miR-1231, miR-328-3p, miR-382-5p*, miR-1207-5p, miR-1301-3p, miR-375, miR-26b-3p*, miR-4505, miR-424-5p, miR-320b, miR-320e, miR-629-3p, miR-1193, miR-142-5p, miR-188-5p, miR-1539, miR-373-3p, mar-1909-5p, miR-378e, miR-15a-3p, miR-324-3p, miR-34c-3p, miR-532-5p, miR-1183, miR-877-3p, miR-124-3p, miR-3120-3p, miR-1237-3p, miR-335-3p, miR-374c-5p
[53]	MicroRNA-155-5p promotes neuroinflammation and central sensitization via inhibiting SIRT1 in a nitroglycerin-induced chronic migraine mouse model	Wen Q.	Nitroglycerin-induced CM mouse model	-	miR-155-5p expression, SIRT1 protein levels;	Increased expression of miR-155-5p and decreased levels of SIRT1 in CM mouse model
[54]	Disease- and headache-specific microRNA signatures and their predicted mRNA targets in peripheral blood mononuclear cells in migraineurs: Role of inflammatory signaling and oxidative stress	Timea Aczél	28 participants (16 with migraine, 12 healthy participants)	N/A ^1^	miRNA of peripheral blood mononuclear cells	miRNAs: hsa-miR-5189-3p (2.59) ^2^, hsa-miR-96-5p (−2.4), hsa-miR-3613-5p (2.55), hsa-miR-99a-3p (2.37), hsa-miR-542-3p (2.4), hsa-miR-6803-3p (2.19), hsa-miR-6731-3p (−2.14), hsa-miR-577 (−2.17), hsa-miR-95-3p (−2.06), hsa-miR-556-3p (−2.18), hsa-miR-412-5p (−2.36), hsa-miR-5701 (−2.24), hsa-miR-3064-5p (2.1), hsa-miR-196a-5p (−2.55), hsa-miR-5189-5p (1.93), hsa-let-7i-3p (−1.82), hsa-miR-1277-5p (2.07), hsa-miR-29b-3p (−1.85), hsa-miR-4676-3p (1.87), hsa-miR-548j-3p (1.91), hsa-miR-1260b (1.78), hsa-miR-326 (1.62), hsa-miR-3174 (1.79), hsa-miR-210-3p (1.77), hsa-miR-32-5p (−1.65), hsa-miR-342-3p (−1.6), hsa-miR-3607-3p (−1.59), hsa-miR-142-5p (−1.54), hsa-miR-192-5p (−1.56), hsa-miR-155-5p (−1.43), and hsa-let-7 g-5p (−1.43)
[55]	Financial stress interacts with *CLOCK* gene to affect migraine	Baksa, D.	2157 participants; 1503 females	N/A	Effect of rs10462028 of *CLOCK* gene on migraine	No direct effect of rs10462028 SNP on migraine; change in miRNA bindings in the 3′UTR of *CLOCK* gene in rs1801260 (G/A): miR-365b-3p G↓, miR-365a-3p G↓, and miR-664a-5p G↓, as well as in rs10462028 (A/G): miR-409-5p A↓
[56]	Circulating microRNAs associated with reversible cerebral vasoconstriction syndrome	Chen S.	30 patients with EM ^4^ during the ictal stage; 30 during the interictal stage; 30 age- and sex-matched HC	Taiwanese	Level of five miRNAs in 30 EM patients during the ictal stage and 30 EM patients during the interictal stage compared with 30 controls	MiR-130a-3p, miR-130b-3p, let-7a-5p, let-7b-5p, and let-7f-5p were investigated; the abundance of let-7a-5p, let-7b-5p, and let-7f-5p was significantly higher in ictal migraine patients compared to that of HC and interictal migraine patients
[57]	Elevated circulating endothelial-specific microRNAs in migraine patients: A pilot study	Cheng C. Y.	30 migraineurs (20 females), 30 age- and sex-matched HC	Taiwanese	miR-155, miR-126, miR-21, and let-7g levels comparison in migraine patients with those in HC level of ICAM-1 (a marker of endothelial dysfunction)	miR-155, miR-126, and let-7g levels were 2–7-fold higher in the interictal migraine patients than in HCs; miR-155, miR-126, and let-7g were positively associated with the level of ICAM-1 in migraine patients
[58]	Neurophysiological and biomolecular effects of erenumab in chronic migraine: An open-label study	De Icco Roberto	40 CM ^5^ patients	N/A	Evaluation of effects of erenumab treatment on the expression levels of miR34a-5p and miR-382-5p.	MiR-382-5p and miR-34a-5p levels were significantly lower after erenumab administration in the overall study population. After erenumab treatment, no significant differences between 30% responder and 30% nonresponder groups were found
[59]	Hsa-miR-34a-5p and hsa-miR-375 as biomarkers for monitoring the effects of drug treatment for migraine pain in children and adolescents: A pilot study	Galleli Luca	24 migraine patients (50% females) without aura in 2 equal groups: treated; untreated 12 age- and sex-matched controls to the untreated group	N/A	Difference in saliva or in blood expression of hsa-miR-34a-5p and hsa-miR-375 between treated group and untreated group.	All enrolled migraineurs constitutively expressed both hsa-miR-34a-5p and hsa-miR-375, without difference with respect to age or gender; decrease of about 50% for hsa-miR-34a-5p and hsa-miR-375 in treated patients compared to untreated patients without difference with respect to age or gender
[60]	MicroRNA profiling in migraine without aura: A pilot study	E. Tafuri	15 females suffering from migraine without aura; 13 HC	N/A	Validation of the following miRNAs by quantitative real-time polymerase chain reaction: miR-22, miR-26a, miR-26b, miR-27b, miR-29b, let-7b, miR-181a, miR-221, miR-30b, miR-30e	miR-27b was significantly upregulated; miR-181a, let-7b, and miR-22 were significantly downregulated
[61]	MiR-30a relieves migraine by degrading CALCA	Y. Zhai	N/A	Chinese	Relationship between miR-30a and *CALCA* in migraine	Expression levels of miR-30a in the peripheral blood of migraine patients were significantly reduced compared with HC
[62]	Differential expression and bioinformatic analysis of the circRNA expression in migraine patients	Jinghan Lin	4 migraine patients, 3 controls	Chinese	Microarray analysis of circRNA of the plasma of migraine patients and healthy controls	2039 circRNAs were detected in patient samples; 794 upregulated, 1245 downregulated relative to controls (fold change ≥ 1:5, *p* < 0:01); the top 10 upregulated circRNAs were hsa_circRNA_103670, hsa_circRNA_101833, hsa_circRNA_103809, hsa_circRNA_104855, hsa_circRNA_104761, hsa_circRNA_102610, hsa_circRNA_103444, hsa_circRNA_100257, hsa_circRNA_103149, and hsa_circRNA_100983; the top 10 downregulated circRNAs were hsa_circRNA_000367, hsa_circRNA_100236, hsa_circRNA_100790, hsa_circRNA_100789, hsa_circRNA_102413, hsa_circRNA_103689, hsa_circRNA_101784, hsa_circRNA_104950, hsa_circRNA_103846, and hsa_circRNA_101698

^1^ Not available; ^2^ fold change; ^3^ healthy controls; ^4^ episodic migraine; ^5^ chronic migraine; ^6^ medication overuse; ^7^ calcitonin gene-related peptide; ↓ a decreased miRNA binding in the 3′UTR of CLOCK gene.

## Data Availability

Not applicable.

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
