# Peer review of "The Epigenetics of Migraine"

_ijms, 2023, doi:10.3390/ijms24119127_

Round 1

Reviewer 1 Report

This is a review about epigenetic findings in relation to migraine.
Please explain all the abbreviations used, including in the abstract.
The methodology must be improved. Please add the number of articles included in your review, the exclusion criteria.
Line 178... correct "migraine307273".
I recommend adding a separate paragraph with the conclusions and to delete this phrase ^This section is not mandatory but can be added to the manuscript if the discussion is unusually long or complex.^
I recommend you to review the article from the technical editing point of view (pay attention to commas and spaces).
The references are appropriate, the article presents 153 references, being up to date. I think that for the first reference it is enough to write only the first 10 authors and et al...instead of a whole page of authors.

Author Response

This is a review about epigenetic findings in relation to migraine.

Answer of the authors:

Thank you for the comment. We are addressing the changes in the following point by point.

1) Please explain all the abbreviations used, including in the abstract.

Thank you for the remark. All abbreviations have been explained accordingly.

2)The methodology must be improved.

The following section has been added to the methodology:

2.1. Literature and search strategy

Search was conducted in the databases PubMed, Scopus, and Google scholar, using the following search terms in different combinations and different long combination chains: (“epigenetics” OR “epigenomics“) AND (“migraine”) AND (“DNA methylation”, OR, “histone acetylation”, OR, “histone methylation”, OR, “CircRNA”,OR , “MicroRNA”).

2.2. Inclusion and Exclusion Criteria

Selected articles were required to meet following criteria:

  • The study contained original data.
  • Represent in vitro or in vivo studies.
  • The study subjects were human or animal.
  • The study written in English.
  • The study describes the interplay between the migraine pathogenesis or potential targets for obtaining effective therapeutic responses and epigenetics.

2.3. Selected studies

27 articles were chosen according to above-mentioned criteria to be presented in this review. (Table 1,2)

3) Please add the number of articles included in your review, the exclusion criteria.

We have added accordingly.

4)Line 178... correct "migraine307273".

We have corrected it accordingly.

5)I recommend adding a separate paragraph with the conclusions and to delete this phrase ^This section is not mandatory but can be added to the manuscript if the discussion is unusually long or complex.^

Thank you for the recommendation. Following section added to the manus:

  1. Conclusion

We conclude that the study of migraine epigenetics is still in a very early stage, but promising data are produced at an increasing speed. At the time of our review, we were able to find very few confirmative studies on this topic. Clinical studies on the migraine epigenetics are characterized by small numbers of participants and inconsistent methodology. Furthermore, results from studies in animal models are difficult to interpret in the context of migraine in humans. These limitations are particularly characteristic for studies on the relationship between epigenetics and the clinical features of the course, severity, and response to therapy of various forms of migraine in different groups of patients, which are important for clinical practice in the era of personalized medicine. The most promising targets for study are methylation of CGRP system genes, such as CALCA and RAMP1. A larger study of epigenetic changes in SH2D5 and NPTX2, COMT, GIT2, ZNF234 and SOCS1 genes is necessary to understand the processes of migraine chronicity and MOH development. Studies of the endocannabinoid system as well as histone acetylation processes and the JNK/c-Jun cascade as potential targets for migraine therapy seem very promising. Studies of microRNAs, especially miR-34a-5p and miR-382-5p, have the potential to shed light on questions of migraine pathophysiology, response to therapy, and chronicity of headache. Larger studies, over longer running times, are needed to confirm the hitherto gathered results.

6)I recommend you to review the article from the technical editing point of view (pay attention to commas and spaces).

Thank you for the remark. We have reviewed the manus accordingly.

7) The references are appropriate, the article presents 153 references, being up to date. I think that for the first reference it is enough to write only the first 10 authors and et al...instead of a whole page of authors.

We have changed it accordingly.

Reviewer 2 Report

In this study, Zobdeh et al. focused on the epigenetic causes of migraine. In particular, they prepared a review on DNA and histone methylation, acetylation and the effects of microRNAs in migraine. While the study is useful because it addresses epigenetic mechanisms, which is an under-studied area of ​​migraine, some points require re-evaluation. First of all, English grammar should be checked, because there are mistakes in many places and therefore it becomes a difficult text to understand. Some of the titles (i.e. title 6) should be more informative. Introduction for methylation and acetylation  (titles 3.1 and 4) is too long and should be shortened. Besides, 'title 4' should be 'title 3.2'. 

There is a massive flow of information throughout the text which sounds very complicated. A reason for this complication may be that the text flow is a bit complex and irregular. The relationship between subsequent titles can not be well understood. Therefore it is difficult for the reader to understand. Making the transitions between consecutive titles a little more obvious by using connecting sentences can provide a more understandable and clear expression.

English grammar should be checked, because there are mistakes in many places and therefore it becomes a difficult text to understand.

Author Response

In this study, Zobdeh et al. focused on the epigenetic causes of migraine. In particular, they prepared a review on DNA and histone methylation, acetylation and the effects of microRNAs in migraine. While the study is useful because it addresses epigenetic mechanisms, which is an under-studied area of ​​migraine, some points require re-evaluation. First of all, English grammar should be checked, because there are mistakes in many places and therefore it becomes a difficult text to understand. Some of the titles (i.e. title 6) should be more informative. Introduction for methylation and acetylation (titles 3.1 and 4) is too long and should be shortened. Besides, 'title 4' should be 'title 3.2'. English grammar should be checked, because there are mistakes in many places and therefore it becomes a difficult text to understand.

The review has been reorganized, adding information in the introduction regarding microRNA and CirRNA section 10 and 11.

Headers have been adapted as for example title 4 changed to 3.2 as per your suggestion.

Grammar and commas and spaces have been rechecked.

There is a massive flow of information throughout the text which sounds very complicated. A reason for this complication may be that the text flow is a bit complex and irregular. The relationship between subsequent titles can not be well understood. Therefore it is difficult for the reader to understand. Making the transitions between consecutive titles a little more obvious by using connecting sentences can provide a more understandable and clear expression.

Thank you for the remark, The sequence of titles has been reorganized and changed to obtain a better flow.

Reviewer 3 Report

The manuscript entitled "The epigenetics of migraine" is a  review of possible molecular markers in migraine.

The review was well written but can be improved with some figures/schemes to illustrate the topics. 

The Methodology must be detailed in how the authors choose the articles cited in this review. I think that there are more than 153 references about the theme "Epigenetic Migraine".

I would like to see also a tentative of:

1) correlation with markers with females, the main gender claimed by authors that are affected by migraine.

2) Any study with drugs used in migraine and their epigenetic influence of then in the markers highlighted in this review.

Minor points:

Line 178: I think that a draft of reference was in th text: "...migraine307273 [31,73 ,74]." 

Line 180: sequence references were cited "different subtypes of a specific disease [75,76,77,78]." Unless the IJMS recommends this format, my suggestion is to format it to "[75-78]".

Line 220 and line 329: the same format suggestion to sequence references.

Secction 14 Conclusions

The text seems to be an instruction from IJMS.

The Section Discussion, in my opinion, lacks conclusions. I would like to see a Conclusion section in a few lines (10-15) highlighting the main points of the manuscript. As an example: what epigenetic factors have real potential in individual therapy?

Author Response

The manuscript entitled "The epigenetics of migraine" is a  review of possible molecular markers in migraine.

Answer of the authors:

Thank you for the comment. We are addressing the changes in the following point by point.

1) The review was well written but can be improved with some figures/schemes to illustrate the topics. 

Figures have been added accordingly.

2) The Methodology must be detailed in how the authors choose the articles cited in this review. I think that there are more than 153 references about the theme "Epigenetic Migraine".

Many thanks for the remark. We would like to point out that we have written a cumulative review and not a systematic review. However, we believe that the reviewer is right that we should provide more information regarding the selection criteria for the studies. In line with this, we have added the following further explaining section to material and methods:

“2.2. Inclusion and Exclusion Criteria

Selected articles were required to meet following criteria:

  • The study contained original data.
  • Represent in vitro or in vivo studies.
  • The study subjects were human or animal.
  • The study written in English.
  • The study describes the interplay between the migraine pathogenesis or potential targets for obtaining effective therapeutic responses and epigenetics.”

I would like to see also a tentative of:

1) correlation with markers with females, the main gender claimed by authors that are affected by migraine.

2) Any study with drugs used in migraine and their epigenetic influence of then in the markers highlighted in this review.

Thank you for the suggestion. Following section have been added to the manuscript:

MicroRNAs and the anti-CGRP agent erenumab. First studies start to investigate drug-induced changes in microRNA expression in migraineurs, which are of particular interest. De Icco Roberto et al. showed that miR-382-5p and miR-34a-5p expression significantly decreased after 84 days of first erenumab administration in a group of 40 migraineurs. However, no statistically significant relationship between the response to treatment and the expression levels of these microRNAs was found [200]. (Figure 2)

MicroRNA and NSAIDs. A study by Galleli et al. included 24 migraineurs (12 treated with acetaminophen or a combination of ibuprofen and magnesium and 12 untreated) and observed a 50% decrease of has-miR-34a-5p and hsa-miR-375 expression in the group of treated patients compared to untreated patients regardless of sex and age [201]. (Figure 2)

Minor points:

Line 178: I think that a draft of reference was in th text: "...migraine307273 [31,73 ,74]." 

We have corrected it accordingly.

Line 180: sequence references were cited "different subtypes of a specific disease [75,76,77,78]." Unless the IJMS recommends this format, my suggestion is to format it to "[75-78]".

Format have been changed accordingly.

Line 220 and line 329: the same format suggestion to sequence references.

Format have been changed accordingly.

Secction 14 Conclusions

The text seems to be an instruction from IJMS.

The Section Discussion, in my opinion, lacks conclusions. I would like to see a Conclusion section in a few lines (10-15) highlighting the main points of the manuscript. As an example: what epigenetic factors have real potential in individual therapy?

Thank you for the recommendation. Following section added to the manus:

  1. Conclusion

We conclude that the study of migraine epigenetics is still in a very early stage, but promising data are produced at an increasing speed. At the time of our review, we were able to find very few confirmative studies on this topic. Clinical studies on the migraine epigenetics are characterized by small numbers of participants and inconsistent methodology. Furthermore, results from studies in animal models are difficult to interpret in the context of migraine in humans. These limitations are particularly characteristic for studies on the relationship between epigenetics and the clinical features of the course, severity, and response to therapy of various forms of migraine in different groups of patients, which are important for clinical practice in the era of personalized medicine. The most promising targets for study are methylation of CGRP system genes, such as CALCA and RAMP1. A larger study of epigenetic changes in SH2D5 and NPTX2, COMT, GIT2, ZNF234 and SOCS1 genes is necessary to understand the processes of migraine chronicity and MOH development. Studies of the endocannabinoid system as well as histone acetylation processes and the JNK/c-Jun cascade as potential targets for migraine therapy seem very promising. Studies of microRNAs, especially miR-34a-5p and miR-382-5p, have the potential to shed light on questions of migraine pathophysiology, response to therapy, and chronicity of headache. Larger studies, over longer running times, are needed to confirm the hitherto gathered results.

Reviewer 4 Report

Paper is good, however, authors needs to clarify these points

Abstract:

1.    Clarify the purpose of the review: The abstract could benefit from a clearer statement of the specific objective or research question that the review aims to address. For instance, the abstract could state whether the review aims to identify epigenetic mechanisms that contribute to migraine pathophysiology or to highlight potential therapeutic targets.

2.    Provide more context on the current state of migraine therapy: While the abstract mentions that therapy success rates for migraine are still unsatisfying, it would be helpful to provide more information on the current state of migraine therapy. This could include a brief overview of existing treatments, their limitations, and the unmet needs of patients.

3.    Emphasize the potential implications of the findings: The abstract mentions that epigenetics could be a promising avenue to discover potential therapeutic targets for migraine treatment and monitoring. However, it would be helpful to elaborate on the potential implications of the findings for patients and clinicians. For example, could the identification of epigenetic biomarkers lead to personalized treatments or improved diagnostic tools?

4.    Provide more information on the discussed genes and microRNA molecules: The abstract lists several genes and microRNA molecules that have been implicated in migraine. Providing more information on these molecules, such as their functions and potential roles in migraine pathophysiology, could help readers better understand the significance of the findings.

5.    Clarify the need for further research: The abstract mentions that more studies are needed to verify the early findings on epigenetic mechanisms in migraine. It would be helpful to explain why further research is necessary, what questions remain unanswered, and what potential barriers exist to advancing the field.

6.    Provide a clear conclusion: The abstract could benefit from a clear concluding statement that summarizes the main findings and their implications. For example, the abstract could state that epigenetic mechanisms appear to have a high potential to contribute to a better understanding of migraine pathophysiology and therapy response, and that further research in this area is necessary to fully realize this potential.

Introduction

1.    Clarify the purpose: It is important to clarify the purpose of the article in the introduction. This helps readers understand what they can expect from the article. In this case, it would be helpful to explain if the article is a review or a research paper.

2.    Simplify language: The language used in the introduction is quite technical, which may make it difficult for some readers to understand. Consider simplifying some of the language to make it more accessible.

3.    Improve sentence structure: Some of the sentences in the introduction are quite long and complex. Consider breaking them up into shorter, more concise sentences to improve readability.

4.    Add context: While the introduction provides a lot of information about migraine, it would be helpful to provide some context around why it is important to study. For example, how does it impact the lives of people who suffer from it, and what are some of the current treatment options?

5.    Provide a clear thesis statement: The introduction should end with a clear thesis statement that outlines the main argument of the article.

Methodology:

1.    Use of additional databases: While PubMed, Scopus, and Google Scholar are popular databases, there are other relevant databases like Embase, Cochrane Library, and Web of Science that can be included to ensure comprehensive coverage of relevant studies.

2.    Use of Boolean operators: The search terms can be combined using Boolean operators (AND, OR, NOT) to refine the search and obtain more relevant studies.

3.    Inclusion and exclusion criteria: Clearly define the inclusion and exclusion criteria for selecting studies to be included in the review. For example, studies that only investigate animal models or studies that do not measure epigenetic changes may be excluded.

4.    Systematic screening: To ensure a comprehensive search, all potentially relevant studies should be screened systematically using pre-defined inclusion and exclusion criteria.

5.    Data extraction: The data extracted from the selected studies should be tabulated in a structured format to facilitate analysis.

Section 3:

1.    Provide examples: Providing examples of how DNA methylation affects gene expression or how mutations or dysregulation of DNMT3 lead to specific pathological conditions could help readers understand the significance of DNA methylation in biological processes.

2.    Link to relevant studies: Adding links to studies that have explored the role of DNA methylation in specific conditions or processes could help readers explore the topic further and understand the practical implications of DNA methylation research

Section 4:

1.    Information on the role of histone acetylation in other biological processes: Histone acetylation is a crucial epigenetic modification that regulates gene expression in various biological processes. You can research and write about the role of histone acetylation in cancer, embryonic development, and immune response, among others needs to be provided

2.    the function of other epigenetic modifications in the brain: While histone acetylation plays a critical role in brain function and neurological disorders, there are other epigenetic modifications that are equally important. You can explore the role of DNA methylation, histone methylation, and non-coding RNA in the brain needs to be added

3.    Role of the potential of HDAC inhibitors as therapeutic agents needs to be provided: HDAC inhibitors have been shown to have promising therapeutic effects in various diseases, including cancer and neurodegenerative disorders. You can research and write about the potential of HDAC inhibitors as therapeutic agents for migraine or other neurological disorders.

4.    Discuss the impact of environmental factors on epigenetic regulation: Environmental factors, such as stress, diet, and exposure to toxins, can influence epigenetic modifications. You can explore the impact of these environmental factors on histone acetylation and other epigenetic modifications in the brain, and how they might contribute to the development of neurological disorders.

5.    What are the ethical implications of epigenetic modification? Epigenetic modifications can have lasting effects on gene expression, which can be passed down from generation to generation. You can examine the ethical implications of epigenetic modification, such as the potential for epigenetic engineering or the responsibility of parents for the epigenetic health of their children.

Section 6:

1.    Investigating role about the epigenetic regulation of the CALCA gene and its potential role in migraine pathogenesis in larger and more diverse patient populations, as well as in animal models.

2.    Some points on studies to further elucidate the role of RAMP1 in migraine pathogenesis, including investigating its epigenetic regulation and potential interactions with CGRP receptors.

3.    Exploring the potential use of epigenetic modifications as therapeutic targets for the treatment of migraine needs to be added

4.    Information on examining other potential genetic and epigenetic factors that may contribute to migraine pathogenesis, such as other neuropeptides, receptors, and signaling pathways.

5.    Information on investigating potential environmental factors that may influence the epigenetic regulation of genes involved in migraine pathogenesis

Section 8:

1.    add information on the epigenetic regulation of the endocannabinoid system (ES) in migraine to determine if other mechanisms, such as DNA hypomethylation and histone hyperacetylation, are involved in gene expression regulation.

3.    Add information whether epigenetic changes in the ES contribute to the development or progression of migraine, as well as other neurological disorders.

4.    potential therapeutic benefits of targeting the ES in the treatment of migraine and other neurological disorders information needs to be added

5.    Comparison on the expression and DNA methylation levels of genes of ES components between different types of migraine, such as chronic migraine with medication overuse and episodic migraine, to determine if there are differences in the epigenetic regulation of the ES needs to be added.

6.     Results of the effects of different interventions, such as pharmacological or lifestyle interventions, on the epigenetic regulation of the ES in migraine and other neurological disorders needs to be added.

FFigures needs to be added.

Minor changes required

Author Response

Paper is good, however, authors needs to clarify these points

Abstract:

  1. Clarify the purpose of the review: The abstract could benefit from a clearer statement of the specific objective or research question that the review aims to address. For instance, the abstract could state whether the review aims to identify epigenetic mechanisms that contribute to migraine pathophysiology or to highlight potential therapeutic targets.

Answer of the authors:

Thank you for the remark. The following section has been added to the abstract:

“In this review, we summarize the state of the art regarding epigenetic findings in relation to migraine pathogenesis and potential therapeutic targets, with a focus on DNA methylation, histone acetylation, and microRNA-dependent regulation.”

  1. Provide more context on the current state of migraine therapy: While the abstract mentions that therapy success rates for migraine are still unsatisfying, it would be helpful to provide more information on the current state of migraine therapy. This could include a brief overview of existing treatments, their limitations, and the unmet needs of patients.

Thank you for the remark. The following section has been added to the abstract:

“A wide range of different drug classes such as triptans, antidepressants, anticonvulsants, analgesics, and beta-blockers are used in acute and preventive migraine therapy. Despite a considerable progress in the development of novel and targeted therapeutic interventions during recent years, e.g., drugs that inhibit the calcitonin gene-related peptide (CGRP) pathway, therapy success rates are still unsatisfying. The diversity in drug classes used in migraine therapy reflects partly the limited perception of migraine pathophysiology.”

  1. Emphasize the potential implications of the findings: The abstract mentions that epigenetics could be a promising avenue to discover potential therapeutic targets for migraine treatment and monitoring. However, it would be helpful to elaborate on the potential implications of the findings for patients and clinicians. For example, could the identification of epigenetic biomarkers lead to personalized treatments or improved diagnostic tools?

Thank you for the remark. The following section has been added to the abstract:

“A better understanding of the causes and consequences of migraine-associated epigenetic changes could help to better understand migraine risk, pathogenesis, development, course, and prognosis. Additionally, it could be a promising avenue to discover new therapeutic targets for migraine treatment and monitoring.”

  1. Provide more information on the discussed genes and microRNA molecules: The abstract lists several genes and microRNA molecules that have been implicated in migraine. Providing more information on these molecules, such as their functions and potential roles in migraine pathophysiology, could help readers better understand the significance of the findings.

Thank you for the remark. The following section has been added to the abstract:

“Several genes and their methylation pattern such as CALCA (migraine symptoms and age of migraine onset), RAMP1, NPTX2 and SH2D5 (migraine chronification) and microRNA molecules such as miR-34a-5p and miR-382-5p (treatment response) seem especially worth to be further studied regarding their role in migraine pathogenesis, course and therapy.”

  1. Clarify the need for further research: The abstract mentions that more studies are needed to verify the early findings on epigenetic mechanisms in migraine. It would be helpful to explain why further research is necessary, what questions remain unanswered, and what potential barriers exist to advancing the field.

Thank you for the remark. The following section has been added to the abstract:

“However, further studies with larger sample sizes are needed to verify these early findings and to be able to establish epigenetic targets as disease predictors or therapeutic targets.”

  1. Provide a clear conclusion: The abstract could benefit from a clear concluding statement that summarizes the main findings and their implications. For example, the abstract could state that epigenetic mechanisms appear to have a high potential to contribute to a better understanding of migraine pathophysiology and therapy response, and that further research in this area is necessary to fully realize this potential.

Thank you for the remark. The following section has been added to the abstract:

“Epigenic changes could be a potential tool to better understanding regarding migraine pathophysiology and identifying new therapeutic approaches. However, further studies with larger sample sizes are needed to verify these early findings and to be able to establish epigenetic targets as disease predictors or therapeutic targets.”

Introduction

  1. Clarify the purpose: It is important to clarify the purpose of the article in the introduction. This helps readers understand what they can expect from the article. In this case, it would be helpful to explain if the article is a review or a research paper.

Thank you for the remark. The following section has been added to the introduction:

“In this review we shed light on the current knowledge regarding the role of DNA methylation, histone acetylation and microRNA shifts in migraine pathophysiology and their potential as future therapeutic predictors or targets which might be considered valuable for further research.”

  1. Simplify language: The language used in the introduction is quite technical, which may make it difficult for some readers to understand. Consider simplifying some of the language to make it more accessible.

Thank you, we have changed the introduction section accordingly.

  1. Improve sentence structure: Some of the sentences in the introduction are quite long and complex. Consider breaking them up into shorter, more concise sentences to improve readability.

We have changed the introduction section accordingly.

  1. Add context: While the introduction provides a lot of information about migraine, it would be helpful to provide some context around why it is important to study. For example, how does it impact the lives of people who suffer from it, and what are some of the current treatment options?

Thank you for the remark. The following section has been added to the introduction:

“Triptans (e.g., sumatriptan, zolmitriptan) and non-steroidal anti-inflammatory drugs (e.g., Ibuprofen) have been the leading options for treating acute migraines for many years. Additionally, the most recent therapeutic agents such as gepants (e.g., Rimegepant) and ditans (Lasmiditan) are considered promising options for the treatment of acute migraine [4,5]. For preventive treatment, different drug classes including beta-blockers (e.g., propranolol), tricyclic antidepressants (e.g., amitriptyline) and anti-convulsants (eg., topiramate) have been used. Furthermore, novel drugs which inhibit calcitonin gene-related peptide (CGRP) or its receptor as validated targets for migraine therapy have been shown to be efficient in migraine preventive treatment. Examples are monoclonal antibodies which either block CGRP (e.g., Galcanezumab) or CGRP receptor (e.g., Erenumab)[5,6]. The diversity in drug classes used for acute and preventive treatment with various mechanisms of actions reflects the overall still limited understanding of migraine pathophysiology.”

  1. Provide a clear thesis statement: The introduction should end with a clear thesis statement that outlines the main argument of the article.

Thank you for the remark. The following section has been added to the introduction part:

“In this review, we shed light on the current knowledge regarding the role of DNA methylation, histone acetylation, and microRNA dysregulation in migraine pathophysiology and their potential as future therapeutic predictors or targets which might be considered valuable for further research.”

Methodology:

  1. Use of additional databases: While PubMed, Scopus, and Google Scholar are popular databases, there are other relevant databases like Embase, Cochrane Library, and Web of Science that can be included to ensure comprehensive coverage of relevant studies.

We have changed accordingly.

  1. Use of Boolean operators: The search terms can be combined using Boolean operators (AND, OR, NOT) to refine the search and obtain more relevant studies.

We have changed accordingly.

“(“epigenetics” OR “epigenomics“) AND (“migraine”) AND (“DNA methylation”, OR, “histone acetylation”, OR, “histone methylation”, OR, “CircRNA”,OR , “MicroRNA”).”

  1. Inclusion and exclusion criteria: Clearly define the inclusion and exclusion criteria for selecting studies to be included in the review. For example, studies that only investigate animal models or studies that do not measure epigenetic changes may be excluded.

Many thanks for the remark. We would like to point out that we have written a cumulative review and not a systematic review. However, we believe that the reviewer is right that we should provide more information regarding the selection criteria for the studies. In line with this, we have added the following further explaining section to material and methods:

“2.2. Inclusion and Exclusion Criteria

Selected articles were required to meet following criteria:

  • The study contained original data.
  • Represent in vitro or in vivo studies.
  • The study subjects were human or animal.
  • The study written in English.
  • The study describes the interplay between the migraine pathogenesis or potential targets for obtaining effective therapeutic responses and epigenetics.”

  1. Systematic screening: To ensure a comprehensive search, all potentially relevant studies should be screened systematically using pre-defined inclusion and exclusion criteria.

Many thanks for the remark, as mentioned above, this is not a systematic review (and was not intended to be from the beginning and not the task in the frame of this invited review), which would need another methodological approach. We have written a cumulative review giving an overview of the state of the art regarding migraine and epigenetics. As mentioned above we have added some more information regarding the selection of the studies included.

  1. Data extraction: The data extracted from the selected studies should be tabulated in a structured format to facilitate analysis.

Thank you for the remark. As clarified in the answer to question 3 and 4 we have updated the methodology part and listed the studies that we identified in tables 1,2 of the results section.

Section 3:

  1. Provide examples: Providing examples of how DNA methylation affects gene expression or how mutations or dysregulation of DNMT3 lead to specific pathological conditions could help readers understand the significance of DNA methylation in biological processes.

Answer of the authors:

Thank you for the remark. We provided now examples of the role of DNA methylation in specific pathological conditions in section 3.1 DNA methylation and demethylation.

  1. Link to relevant studies: Adding links to studies that have explored the role of DNA methylation in specific conditions or processes could help readers explore the topic further and understand the practical implications of DNA methylation research

We have introduced links now, as suggested.

Section 4:

  1. Information on the role of histone acetylation in other biological processes: Histone acetylation is a crucial epigenetic modification that regulates gene expression in various biological processes. You can research and write about the role of histone acetylation in cancer, embryonic development, and immune response, among others needs to be provided

Answer of the authors:

Thank you for the remark. We added now information on the role of histone acetylation in cancer, immune response and embryonic development and summarized this information in the sections 3.2.2, 3.2.3, and 3.2.4 respectively.

  1. the function of other epigenetic modifications in the brain: While histone acetylation plays a critical role in brain function and neurological disorders, there are other epigenetic modifications that are equally important. You can explore the role of DNA methylation, histone methylation, and non-coding RNA in the brain needs to be added

Answer of the authors:

Thank you for the remark. We added examples of disorders with altered DNA methylation and a brief histone methylation section.

  1. Role of the potential of HDAC inhibitors as therapeutic agents needs to be provided: HDAC inhibitors have been shown to have promising therapeutic effects in various diseases, including cancer and neurodegenerative disorders. You can research and write about the potential of HDAC inhibitors as therapeutic agents for migraine or other neurological disorders.

Answer of the authors:

Thank you for the remark. In line with your request, we added an additional section 3.2.5 to the review with the title “Histone deacetylase inhibitors as therapeutic agents in neurologic disorders”.

  1. Discuss the impact of environmental factors on epigenetic regulation: Environmental factors, such as stress, diet, and exposure to toxins, can influence epigenetic modifications. You can explore the impact of these environmental factors on histone acetylation and other epigenetic modifications in the brain, and how they might contribute to the development of neurological disorders.

Answer of the authors:

Thank you for the remark. In line with your suggestion, we added a new section 4. “Environment and epigenetics, dedicated to the impact of environmental factors on epigenetic regulation.”

  1. What are the ethical implications of epigenetic modification? Epigenetic modifications can have lasting effects on gene expression, which can be passed down from generation to generation. You can examine the ethical implications of epigenetic modification, such as the potential for epigenetic engineering or the responsibility of parents for the epigenetic health of their children.

Answer of the authors:

Thank you for the remark. The following section has been added to the discussion part:

“With the growth in knowledge of the role of epigenetics in migraine pathophysiology and treatment, the field of headache research will be soon confronted also with putative ethical implications that a rapidly developing field such as epigenetics has to deal with it. The possibility of epigenetic engineering, which involves the intentional alteration of the epigenome in order to achieve desired outcomes, such as disease prevention or the augmentation of certain disease traits, needs to be carefully balanced and evaluated regarding the risk-benefit and social and economic implications in association with the accessibility of the technology. Other ethical issues of epigenetics comprise the possibility to prevent certain epigenetic heritable triggers of a disease due to a preventive parental behavior, e.g., through the adaptation of certain lifestyle factors that may be also relevant topics for migraine in the future when the disorder is better epigenetically understood. [205,206,207].”

Section 6:

  1. Investigating role about the epigenetic regulation of the CALCA gene and its potential role in migraine pathogenesis in larger and more diverse patient populations, as well as in animal models.

Answer of the authors:

Thank you for the remark. To the best of our knowledge, all available articles investigating the epigenetic regulation of the CALCA gene and its potential role in migraine pathogenesis in patients or in animal models were included in our review.

  1. Some points on studies to further elucidate the role of RAMP1 in migraine pathogenesis, including investigating its epigenetic regulation and potential interactions with CGRP receptors.

Answer of the authors:

Thank you for the remark. We added now more information on the epigenetic regulation of RAMP1 and the interaction of RAMP1 with CGRP in section 8.1.2.

  1. Exploring the potential use of epigenetic modifications as therapeutic targets for the treatment of migraine needs to be added

Answer of the authors:

Thank you for the remark. We added a separate section to briefly explore this topic:

“9. Epigenetics as a therapeutic target in migraine treatment

Based on the first evidence that epigenetics plays a role in migraine it can be assumed that epigenetics may also be valuable as a therapeutic target in migraine. A number of epigenetic therapeutics are approved for the treatment of various cancers, such as myelodysplastic syndrome, certain types of leukemia, large B-cell lymphoma, and cutaneous T-cell lymphomas, and even more are currently developed for cancer treatment. Of the nine epigenetic therapeutics approved by the U.S. Food and Drugs Administration, two are DNMT inhibitors (DNMTIs), four are the aforementioned HDACs, two are isocitrate dehydrogenase inhibitors (IDHs), and one is the EZH2 inhibitor tazemetostat. Generally, anti-cancer drugs have been developed that target DNMTs, HATs, HDACs, HDMs, HMTs, and IDHs. IDHs inhibit TET enzymes. Thus, IDH inhibitors relieve TET enzyme inhibition, leading to anti-cancer effects [177]. However, in the case of migraine, the relationship between specific epigenetic targets and the pathophysiology of migraine needs to be further established in order to develop novel epigenetic therapeutic agents.”

  1. Information on examining other potential genetic and epigenetic factors that may contribute to migraine pathogenesis, such as other neuropeptides, receptors, and signaling pathways.

Answer of the authors:

Thank you for the remark. To the best of our knowledge, all available literature on epigenetic factors and migraine was used in the present review.

We added a brief section 8.3, discussing other genetic factors that may contribute to migraine pathogenesis.

  1. Information on investigating potential environmental factors that may influence the epigenetic regulation of genes involved in migraine pathogenesis

Answer of the authors:

Thank you for the remark. We found studies investigating the influence of stress on the epigenetic regulation of ES. We added this information in section 8.2.2 General aspects of the epigenetic regulation of the endocannabinoid system.

Section 8:

  1. add information on the epigenetic regulation of the endocannabinoid system (ES) in migraine to determine if other mechanisms, such as DNA hypomethylation and histone hyperacetylation, are involved in gene expression regulation.

Answer of the authors:

Thank you for the remark. We added the sections 8.2.2, 8.2.3, and 8.2.4 now to our review to explore this topic further.

  1. Add information whether epigenetic changes in the ES contribute to the development or progression of migraine, as well as other neurological disorders.

 Answer of the authors:

Thank you for the remark. We added now information on the role of epigenetic changes in the ES in Alzheimer’s disease in section 8.2.3. We also explored epigenetic changes in the ES in migraineurs in section 8.2.3.

  1. potential therapeutic benefits of targeting the ES in the treatment of migraine and other neurological disorders information needs to be added

Answer of the authors:

Thank you for the remark. We added information on the ability of ES to regulate epigenetics and its potential role as a therapeutic target in section 8.2.4. The endocannabinoid system influences epigenetics.

  1. Comparison on the expression and DNA methylation levels of genes of ES components between different types of migraine, such as chronic migraine with medication overuse and episodic migraine, to determine if there are differences in the epigenetic regulation of the ES needs to be added.

Answer of the authors:

Thank you for the remark. To the best of our knowledge, only one study analyzed ES epigenetics in migraineurs and was included this study in our review.

  1. Results of the effects of different interventions, such as pharmacological or lifestyle interventions, on the epigenetic regulation of the ES in migraine and other neurological disorders needs to be added.

Answer of the authors:

Thank you for the remark. We have rechecked this aspect. To the best of our knowledge, to date, no article has investigated the impact of therapeutic or lifestyle interventions on the epigenetics of ES.

Figures needs to be added.

Figures have been added accordingly.

Reviewer 5 Report

The current manuscript entitled: The epigenetics of migraine by Farzin Zobdeh and colleges has described the epigenetic mediated gene expression could be a promising avenue to discover potential therapeutic targets for migraine treatment and monitoring. Here in this review, the authors summarized the role of DNA methylation, histone acetylation and microRNA dependent regulation in migraine disorder. The authors have done a good job in summarizing the story with appropriate literature survey. The manuscript written in good English and will attract many readers. I recommend this article to be publish in IJMS. 

The authors have written  this MS in good english 

Author Response

The current manuscript entitled: The epigenetics of migraine by Farzin Zobdeh and colleges has described the epigenetic mediated gene expression could be a promising avenue to discover potential therapeutic targets for migraine treatment and monitoring. Here in this review, the authors summarized the role of DNA methylation, histone acetylation and microRNA dependent regulation in migraine disorder. The authors have done a good job in summarizing the story with appropriate literature survey. The manuscript written in good English and will attract many readers. I recommend this article to be publish in IJMS. 

Many thanks for the nice evaluation.

Reviewer 6 Report

This review article, "The epigenetics of migraine," by Farzin Zobdeh and co-author, summarizes the early findings regarding epigenetic results concerning migraine, focusing on DNA methylation, histone acetylation and microRNA-dependent regulation. Several proteins and microRNA molecules have also been discussed. The entire report is well-informed, nicely gathered, analyzed and extracted for a broad range of audiences. Precise structural presentation could benefit researchers in sections 3 to 9. A schematic diagram in the introduction could improve the presentation quality. The present manuscript can be recommended for publication after those modifications.  

Moderate editing of English language to improve the easy readability.

Author Response

This review article, "The epigenetics of migraine," by Farzin Zobdeh and co-author, summarizes the early findings regarding epigenetic results concerning migraine, focusing on DNA methylation, histone acetylation and microRNA-dependent regulation. Several proteins and microRNA molecules have also been discussed. The entire report is well-informed, nicely gathered, analyzed and extracted for a broad range of audiences. Precise structural presentation could benefit researchers in sections 3 to 9. A schematic diagram in the introduction could improve the presentation quality. The present manuscript can be recommended for publication after those modifications. 

Many thanks for the nice evaluation. We have added now 2 figures and we have changed sequence of titles throughout manuscript based on your suggestion. As we have not written a systematic review we have not added a low chart as normally done for systematic reviews. However, based on your suggestion and the suggestion of other reviewers we have further adapted the methods section, stating more precisely the inclusion and exclusion on which the final selection is based.

Round 2

Reviewer 3 Report

The manuscript was improved addressing all issues.

Reviewer 4 Report

addressed all the comments satisfactorily

Very minor corrections are required